# Neural signatures of auditory hypersensitivity following acoustic trauma

**Matthew McGill**[1,2]*, **Ariel E Hight**[1,2]†, **Yurika L Watanabe**[1], **Aravindakshan Parthasarathy**[1,3]‡, **Dongqin Cai**[1,3], **Kameron Clayton**[1,3], **Kenneth E Hancock**[1,3], **Anne Takesian**[1,3], **Sharon G Kujawa**[1,3], **Daniel B Polley**[1,3]

[1]Eaton-Peabody Laboratories, Massachusetts Eye and Ear Infirmary, Boston, United States; [2]Division of Medical Sciences, Harvard Medical School, Boston, United States; [3]Department of Otolaryngology - Head and Neck Surgery, Harvard Medical School, Boston, United States

**\*For correspondence:**
mmcgill@g.harvard.edu

**Present address:** †Department of Otolaryngology-Head and Neck Surgery, New York University School of Medicine, New York, United States; ‡Department of Communication Science and Disorders, University of Pittsburgh, Pittsburgh, United States

**Competing interest:** The authors declare that no competing interests exist.

**Abstract** Neurons in sensory cortex exhibit a remarkable capacity to maintain stable firing rates despite large fluctuations in afferent activity levels. However, sudden peripheral deafferentation in adulthood can trigger an excessive, non-homeostatic cortical compensatory response that may underlie perceptual disorders including sensory hypersensitivity, phantom limb pain, and tinnitus. Here, we show that mice with noise-induced damage of the high-frequency cochlear base were behaviorally hypersensitive to spared mid-frequency tones and to direct optogenetic stimulation of auditory thalamocortical neurons. Chronic two-photon calcium imaging from ACtx pyramidal neurons (PyrNs) revealed an initial stage of spatially diffuse hyperactivity, hyper-correlation, and auditory hyperresponsivity that consolidated around deafferented map regions three or more days after acoustic trauma. Deafferented PyrN ensembles also displayed hypersensitive decoding of spared mid-frequency tones that mirrored behavioral hypersensitivity, suggesting that non-homeostatic regulation of cortical sound intensity coding following sensorineural loss may be an underlying source of auditory hypersensitivity. Excess cortical response gain after acoustic trauma was expressed heterogeneously among individual PyrNs, yet 40% of this variability could be accounted for by each cell's baseline response properties prior to acoustic trauma. PyrNs with initially high spontaneous activity and gradual monotonic intensity growth functions were more likely to exhibit non-homeostatic excess gain after acoustic trauma. This suggests that while cortical gain changes are triggered by reduced bottom-up afferent input, their subsequent stabilization is also shaped by their local circuit milieu, where indicators of reduced inhibition can presage pathological hyperactivity following sensorineural hearing loss.

## Editor's evaluation

This is an important and methodologically compelling paper that provides novel views on the functional plasticity of cortical networks following noise trauma. The combination of cortical recordings, optogenetics, and behavior provides valuable mechanistic insights into the processes that may underlie auditory hypersensitivity after cochlear damage. The extensive and well-illustrated manuscript provides an excellent example of a study in which modern neurophysiology techniques advance comprehension of pathologies related to maladaptive changes in the brain.

## Introduction

Sensory disorder research typically focuses on the mechanisms underlying inherited or acquired forms of sensory loss. But some of the most common and debilitating sensory disorder phenotypes reflect just the opposite problem; not what cannot be perceived, but what cannot *stop* being perceived. Sensory overload typically presents as three inter-related features: (1) an irrepressible perception of phantom stimuli that have no physical environmental source, (2) a selective inability to perceptually suppress distracting sensory features, and (3) the perception that moderate stimuli are uncomfortably intense, distressing, or even painful. One or more of these phenotypes are commonly observed in heterogeneous neurological and psychiatric disorders including autism spectrum disorder (*Klintwall et al., 2011*; *Robertson and Baron-Cohen, 2017*), post-traumatic stress disorder (*Ehlers and Clark, 2000*; *Garfinkel and Liberzon, 2009*), fibromyalgia (*Nielsen and Henriksson, 2007*; *Yunus, 2007*), schizophrenia (*González-Rodríguez et al., 2021*; *Luck and Gold, 2008*), traumatic brain injury (*Nampiaparampil, 2008*), sensorineural hearing loss (SNHL) (*Auerbach et al., 2014*; *Hébert et al., 2013*; *Pienkowski et al., 2014a*), attention deficit hyperactivity disorder (*Ghanizadeh, 2011*), migraine (*Goadsby et al., 2017*), and also as a consequence of normal aging (*Herrmann and Butler, 2021a*).

Although overload phenotypes are reported across many sensory modalities, they are most common and debilitating in hearing, where an estimated 14% of adult population continuously perceives a phantom ringing or buzzing sound (i.e., tinnitus) (*Shargorodsky et al., 2010*), 9% of adults report hypersensitivity, distress, or even pain in response to ordinary environmental sounds (i.e., hyperacusis) (*Pienkowski et al., 2014a*; *Pienkowski et al., 2014b*), and 9% of adults seek health care for poor hearing in complex listening environments but do not have hearing loss (*Parthasarathy et al., 2020a*). While each facet of auditory overload can occur without evidence of peripheral dysfunction, they are common in persons with SNHL arising from age- or noise-induced degeneration of cochlear hair cells and cochlear afferent nerve terminals (for review see *Auerbach and Gritton, 2022*; *Herrmann and Butler, 2021b*; *Noreña, 2011*; *Zeng, 2013*).

In developing sensory systems, deprivation of peripheral input is registered by central sensory neurons via decreased cytosolic calcium levels, which triggers a cascade of genetic, epigenetic, post-transcriptional, and post-translational processes that collectively adjust the electrical excitability of neurons to restore activity back to the baseline set point range (for review see *Harris and Rubel, 2006*; *Turrigiano, 2012*). Compensatory changes in the central auditory pathway can be grouped into three categories: (1) synaptic sensitization via upregulation of receptors for excitatory neurotransmitters (e.g., AMPA receptor scaling) (*Balaram et al., 2019*; *Kotak et al., 2005*; *Sturm et al., 2017*; *Teichert et al., 2017*); (2) synaptic disinhibition via removal of inhibitory neurotransmitter receptors (*Balaram et al., 2019*; *Richardson et al., 2013*; *Sanes and Kotak, 2011*; *Sarro et al., 2008*; *Sturm et al., 2017*); (3) increased intrinsic excitability via changes in the amount and subunit composition of voltage-gated ion channels that set the resting membrane potential, membrane resistance, and spike 'burstiness' (*Li et al., 2015*; *Li et al., 2013*; *Pilati et al., 2012*; *Wu et al., 2016*; *Yang et al., 2012*). Activity changes arising from these compensatory processes are often described as homeostatic plasticity (*Herrmann and Butler, 2021b*; *Schaette and Kempter, 2006*), but in the context of adult cortical plasticity after hearing loss they typically culminate in a failure to maintain neural excitability at a stable set point. Whereas homeostatic changes – by definition – restore neural activity to a baseline activity rate following a perturbation (*Turrigiano, 2012*), neural gain adjustments following adult-onset hearing loss often over-shoot the mark, producing catastrophic downstream consequences at the level of network excitability and sound perception (*Eggermont, 2017*; *Nahmani and Turrigiano, 2014*; *Noreña, 2011*).

Pinpointing the perceptual consequences of excess central auditory gain requires translating molecular and synaptic changes into measurements that can be made in intact, and even behaving animals. With conventional microelectrode recordings, the synaptic and intrinsic compensatory processes described above manifest as increased spontaneous activity rates, increased spike synchrony, steeper slopes of sound intensity growth functions, and poor adaptation to background noise sources (for recent review see *Auerbach and Gritton, 2022*; *Herrmann and Butler, 2021b*). The connection between these extracellular signatures of excess neural gain and auditory perceptual disorders has remained obscure, in part due to a frequent reliance on involuntary behaviors with a relationship to sound perception (for review see *Boyen et al., 2015*; *Brozoski and Bauer, 2016*; *Campolo et al.,*

*2013*; *Hayes et al., 2014*). Further, in vivo measurements have generally taken the form of acute recordings of local field potentials or unidentified excitatory and inhibitory unit types, often in anesthetized animals, and often without detailed measurements of the peripheral insult or the topographic correspondence between neural recording sites and deafferented map regions.

Here, we developed an approach to make more direct operant behavioral measurements of auditory hypersensitivity in mice with detailed characterizations of cochlear lesions. These behavioral measurements were combined with an optical approach to visualize spontaneous and sound-evoked calcium transients in awake mice from hundreds of individual excitatory neurons spanning the entire topographic map of the primary auditory cortex (A1). By performing these measurements before and after noise-induced SNHL, we were able to return to the same neurons over a 3- to 4-week period to identify baseline response features that could predict whether a given neuron would subsequently exhibit stable, homeostatic activity regulation or non-homeostatic excess gain following acoustic trauma.

## Results

### Perceptual hypersensitivity following noise-induced high-frequency SNHL

In humans, a steeply sloping high-frequency hearing loss is a telltale signature of SNHL (*Allen and Eddins, 2010*; *Hannula et al., 2011*). We reviewed 132,504 case records from visitors to the audiology clinic at our institution and determined that 23% of pure tone audiograms fit the description of high-frequency SNHL (*Figure 1A*), underscoring that it is a common clinical condition commonly related to tinnitus, abnormal loudness growth, and poor speech intelligibility in noise (*Horwitz et al., 2002*; *Lewis et al., 2020*; *Moore et al., 1999*; *Oxenham and Bacon, 2003*; *Strelcyk and Dau, 2009*). To model this hearing loss profile in genetically tractable laboratory mice, we induced SNHL through exposure to narrow-band high-frequency noise (16–32 kHz) at 103 dB SPL for 2 hr. Repeated cochlear function testing before and after noise exposure revealed a sustained elevation of high-frequency thresholds for wave 1 of the auditory brainstem response (ABR) and cochlear distortion product otoacoustic emissions (DPOAEs) with mild or no threshold elevation below 16 kHz (*Figure 1B, C*).

Post-mortem cochlear histopathology performed 21 days after noise exposure suggested an anatomical substrate for cochlear function changes due to acoustic trauma. Compared to age-matched unexposed control ears, high-frequency noise exposure eliminated approximately 50% of synaptic contacts between inner hair cells (IHCs) and primary cochlear afferent neurons in the high-frequency base of the cochlea (*Figure 1D*). Noise-induced outer hair cell (OHC) death was only observed in the high-frequency extreme of the cochlear base (*Figure 1E*), though more subtle OHC stereocilia damage was evident throughout mid- and high-frequency cochlear regions (*Figure 1F, G*).

To make a more direct comparison to clinical determinations of hearing loss in humans via pure tone behavioral thresholds, we also performed behavioral tone detection in head-fixed mice (*Figure 1H*). Mice were trained to report their detection of mid- or high-frequency tones (8 and 32 kHz, respectively) by licking a water reward spout shortly after tone delivery. Behavioral thresholds were determined with a modified 2-up, 1-down method-of-limits that presented a combination of liminal tone intensities along with no-tone catch trials (*Figure 1I*). Behavioral detection thresholds were measured every 1–3 days before and after noise-induced SNHL (trauma, $N = 7$) and in control mice exposed to an innocuous noise level (sham, $N = 6$), revealing a stable 45 dB increase in high-frequency tone threshold after traumatic noise exposure without commensurate changes in false alarm rates or mid-frequency detection thresholds (*Figure 1J, K*, statistical analyses are provided in figure legends).

### Behavioral hypersensitivity to cochlear lesion edge frequencies or direct auditory thalamocortical activation after SNHL

High-frequency OHC damage was the likely source of elevated ABR (*Figure 1B*), DPOAE (*Figure 1C*), and behavioral detection (*Figure 1K*) thresholds. The additional loss of auditory nerve afferent synapses onto IHCs (*Figure 1D*) would not be expected to affect hearing thresholds, but could have other influences on sound perception (*Chambers et al., 2016a*; *Henry and Abrams, 2021*; *Lobarinas et al., 2013*; *Resnik and Polley, 2021*). For example, we recently reported that human subjects with normal hearing thresholds but asymmetric degeneration of the left and right auditory nerve perceive

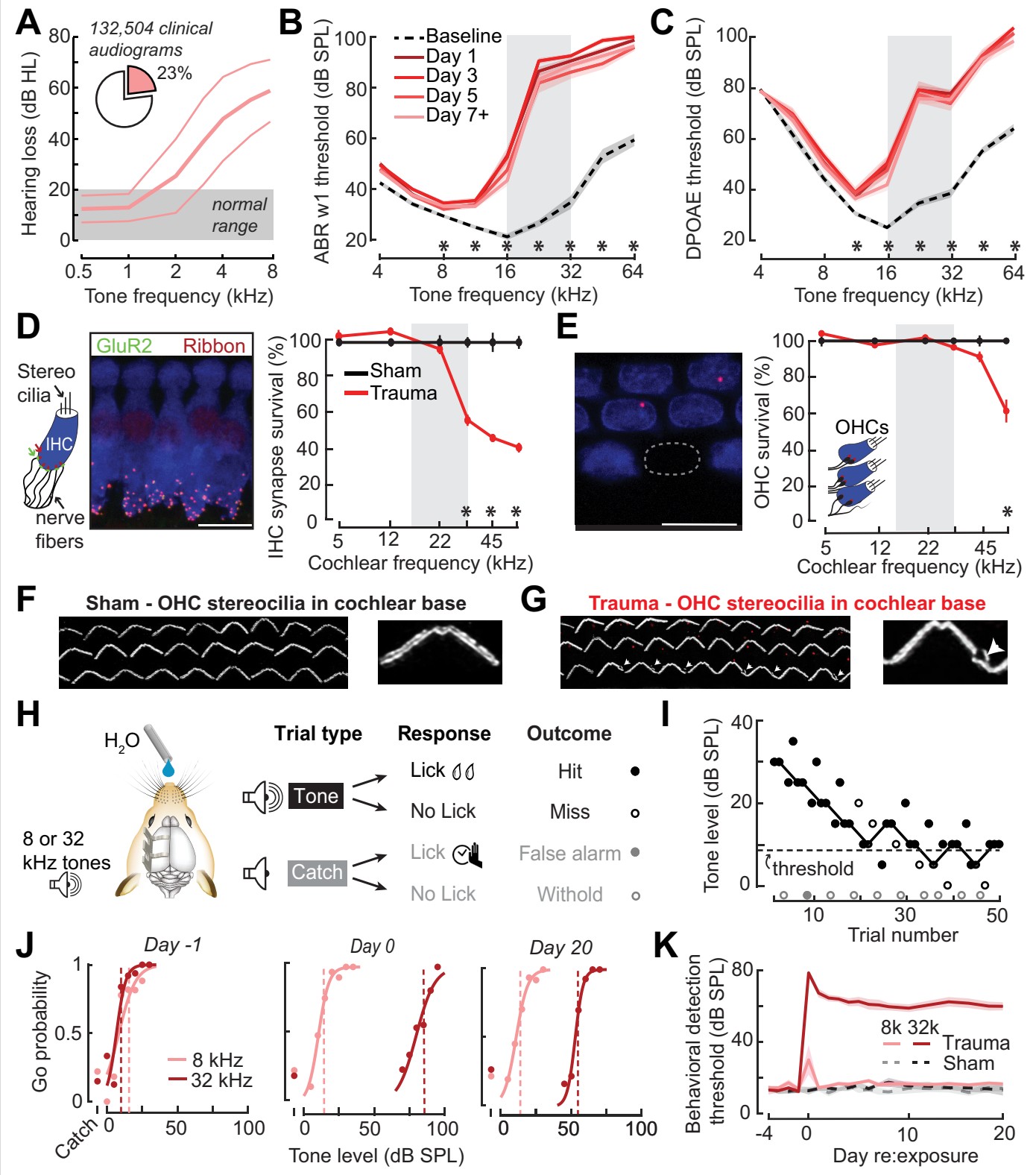

**Figure 1.** Electrophysiological, anatomical, and behavioral confirmation of noise-induced, high-frequency sensorineural hearing loss. (**A**) In human subjects, analysis of 132,504 pure tone audiograms indicates that 23% of visitors to our audiology clinic present with steeply sloping high-frequency hearing loss. Values represent mean ± standard deviation (SD) hearing thresholds in units of dB hearing loss (HL). In mice, response thresholds for wave 1 of the auditory brainstem response (ABR) (**B**) and cochlear distortion product otoacoustic emission (DPOAE) (**C**) measured before and at various

*Figure 1 continued on next page*

*Figure 1 continued*

time points after acoustic trauma show a permanent threshold shift at high frequencies (two-way repeated measures analyses of variance (ANOVAs), Frequency × Time interaction terms for ABR wave 1 [$F$ = 87.51, p = 6 × 10$^{-26}$] and DPOAE [$F$ = 46.44, p = 2 × 10$^{-29}$]). Asterisks denote significant differences between baseline and day 7 + measurements with post hoc pairwise comparisons (p < 0.05). (**D**) Cochlea immunostained for anti-CtBP2 and anti-GluR2a reveal reduced presynaptic ribbon and post-synaptic glutamate receptor patches, respectively, at high-frequency regions of the cochlea (>22 kHz) in trauma mice compared to sham exposure controls (mixed model ANOVA with Group as a factor, Frequency as a repeated measure, and cochlear synapse count as the outcome measure: Group × Frequency interaction term, $F$ = 22.33, p = 4 × 10$^{-11}$). Asterisks denote significant differences between Sham and Trauma with post hoc pairwise comparisons (p < 0.05). (**E**) Loss of outer hair cell (OHC) bodies is limited to the extreme basal regions of the cochlea in noise-exposed animals (Group × Frequency interaction term, $F$ = 11.54, p = 4 × 10$^{-7}$). (**F–G**) Anatomical substrates for cochlear threshold shifts (**B, C**) in more apical cochlear regions can be linked to comparatively subtle OHC stereocilia damage, as visualized by anti-Espin immunolabeling of actin bundle proteins. Cochlear location is approximately 32 kHz. Scale bars represent 10 μm. (**H**) Schematic depicts the design of a head-fixed Go/NoGo tone detection task. (**I**) A modified 2-up, 1-down adaptive staircasing approach to study tone detection thresholds. Example data show one run of 8 kHz tone detection, which finishes at six reversals. (**J**) Logistical fits of 8- and 32 kHz Go probability functions for one mouse measured before, hours after, and 20 days following acoustic trauma. Dotted lines show threshold as determined by the adaptive tracking method. (**K**) Daily behavioral threshold measurements from 13 mice ($N$ = 7 trauma) over an approximate 3-week time period shows a permanent increase in 32 kHz threshold but not 8 kHz after acoustic trauma. Mixed model ANOVA with Group as a factor and both Frequency and Time as repeated measures, main effects for Group [$F$ = 157.76, p = 8 × 10$^{-8}$], Frequency [$F$ = 368.87, p = 9 × 10$^{-10}$], and Time [$F$ = 44.21, p = 6 × 10$^{-53}$], Group × Frequency × Time interaction [$F$ = 37.98, p = 2 × 10$^{-48}$].

tones of fixed physical intensity as louder in the ear with poor auditory nerve integrity, particularly for low intensities near sensation level (*Jahn et al., 2022*). To determine whether mice may be showing evidence of auditory hypersensitivity, we performed a closer inspection of the mouse psychometric detection functions for the spared 8 kHz tone. This analysis confirmed that, following acoustic trauma, the behavioral sensitivity index (*d*-prime, or *d'*) grew more steeply for sound intensities near threshold compared both to pre-exposure baseline detection functions and sham-exposed controls (*Figure 2A, B*). These effects were consistent at the level of individual mice, where we noted an increase in psychometric growth slopes by at least 30% from baseline in 7/7 acoustic trauma mice but changes of this magnitude were not observed in any sham-exposed control mice (*Figure 2B*, right).

Persons with SNHL commonly report loudness recruitment, a disproportionately steep growth of loudness with increasing sound level, that has been accounted for by altered basilar membrane mechanics after OHC damage (*Oxenham and Bacon, 2003*). However, loudness recruitment is most pronounced for frequencies within the range of hearing loss and at intensities well above sensation level (*Buus and Florentine, 2002*; *Moore, 2004*). Here, we observed steeper growth of auditory sensitivity for a spared frequency and at intensities close to sensation level, neither of which would be expected of a purely peripheral origin.

To investigate the possibility that increased neural gain in the central pathway could contribute to auditory hypersensitivity, we built upon pioneering work that directly stimulated deafferented central auditory nuclei to demonstrate a direct association between neural and behavioral hypersensitivity (*Gerken, 1979*). Here, we interleaved acoustic and optogenetic stimulation (*Guo et al., 2015*) to test the hypothesis that mice with high-frequency SNHL would be hypersensitive to stimulation that bypassed the peripheral damage and directly stimulated feedforward excitatory projection neurons in the central auditory pathway. We used an intersectional virus strategy to express channelrhodopsin in auditory thalamocortical neurons, an exclusively glutamatergic feedforward projection system (*Hackett et al., 2016*; *Figure 2C*). Using a Go/NoGo operant task, detection probability was tested in alternating blocks for high-frequency bandpass noise (centered at 32 kHz) or optogenetic thalamocortical stimulation. Following acoustic trauma, high-frequency detection thresholds were elevated by approximately 50 dB ($N$ = 6), whereas sham exposure had no comparable effect ($N$ = 3; *Figure 2D*). Importantly, testing in interleaved trials revealed behavioral hypersensitivity to direct stimulation of auditory thalamocortical neurons after acoustic trauma (*Figure 2E*), as evidenced by significantly increased *d'* at a fixed set of optogenetic stimulation intensities compared to sham controls (*Figure 2F*). Although acoustic trauma can introduce changes in sound processing throughout all stages of the central pathway, the observation that behavioral hypersensitivity was observed to stimulation of thalamocortical projection neurons suggests that compensatory plasticity within the ACtx would be a reasonable place to investigate the neural underpinnings of auditory hypersensitivity following acoustic trauma.

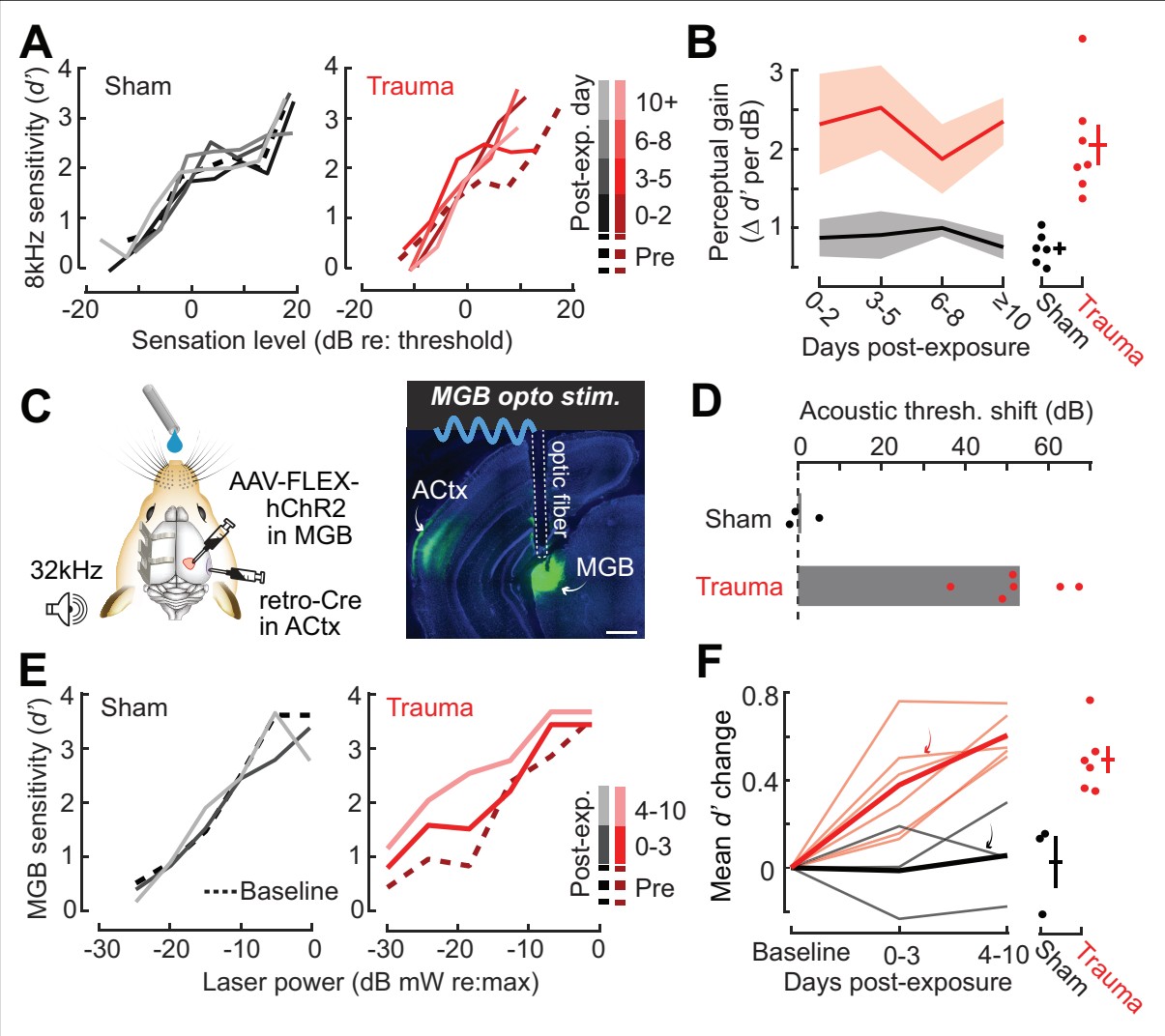

**Figure 2.** Hypersensitivity to sound and direct auditory thalamocortical stimulation following acoustic trauma. (**A**) Behavioral detection functions (reported as the sensitivity index) (*d'*) across time for sham- and noise-exposed example mice show hypersensitivity to spared, mid-frequency tones. (**B**) *Left*, change in perceptual gain for the 8 kHz tone, measured as the mean increase in *d'* per dB increase in sound level, relative to baseline performance. Perceptual gain for an 8 kHz tone is increased in acoustic trauma mice (*N* = 7) compared to sham (*N* = 6) but does not change over post-exposure time (analysis of variance [ANOVA] with Group as a factor and Time as a repeated measure, main effect for Group, *F* = 12.42, p = 0.005; main effect for Time, *F* = 0.52, p = 0.67). *Right*, mean perceptual gain change collapsed across time is separately plotted for each mouse. (**C**) *Left*, preparation for the mixed modality optogenetic and tone detection task. *Right*, ChR2 expression in auditory thalamocortical neurons and placement of the implanted optic fiber relative to retrogradely labeled cell bodies in the medial geniculate body (MGB). (**D**) During 32 kHz tone detection blocks, detection thresholds are elevated by approximately 50 dB following trauma (*N* = 6, paired *t*-test, p = 0.0001) but no change was noted in sham-exposed mice (*N* = 3, p = 0.74). (**E**) Psychometric detection functions for optogenetic auditory thalamocortical stimulation before and after acoustic trauma or sham exposure in two example mice. (**F**) *Left*, summed change across the *d'* function relative to baseline, for noise- and sham-exposed mice. Individual mice are thin lines, group means are thick lines. Example mice from E are indicated by the arrows. After trauma, mice became hypersensitive to MGB stimulation, suggesting an auditory thalamocortical contribution to perceptual hypersensitivity (ANOVA with Group as a factor and post-exposure Time as a repeated measure; main effect for Group, *F* = 15.54, p = 0.006; main effect for time, *F* = 4.65, p = 0.07). *Right*, *d'* collapsed across time is separately plotted for each mouse.

## Chronic imaging in A1 reveals tonotopic remapping and dynamic spatiotemporal adjustments in neural response gain after acoustic trauma

Following topographically restricted cochlear lesions, neurons in deafferented ACtx map regions reorganize to preferentially encode spared cochlear frequencies bordering the damaged region

without accompanying elevations in response threshold (*Engineer et al., 2011*; *Noreña et al., 2003*; *Robertson and Irvine, 1989*; *Seki and Eggermont, 2003*; *Yang et al., 2011*). Because the degree and form of cortical plasticity following cochlear deafferentation can differ between excitatory neurons and various types of inhibitory neurons (*Masri et al., 2021*; *Resnik and Polley, 2021*; *Wang et al., 2022*), we restricted ACtx activity measurements to excitatory neurons via chronic calcium imaging in Thy1-GCaMP6s × CBA mice, where expression of a high-sensitivity genetically encoded calcium indicator is limited to cortical PyrNs (*Chen et al., 2013*; *Romero et al., 2019*). In initial experiments, we performed widefield epifluorescence calcium imaging in awake mice, which offers simultaneous visualization of all fields within the ACtx at mesoscale resolution (*Figure 3A*). These experiments confirmed that tonotopic maps were relatively stable over ~4 weeks in mice with normal hearing experience (*Issa et al., 2014*; *Romero et al., 2019*) but underwent large-scale reorganization throughout the week following acoustic trauma before stabilizing in a state that over-represented 8–16 kHz tones that bordered the high-frequency cochlear lesion (*Figure 3B*).

Having confirmed that A1 was a promising target for more detailed characterizations, we changed our approach to two-photon cellular-scale measurements of A1 layer 2/3 neurons (*Figure 3C*), a cortical layer that shows robust central gain enhancements after acoustic trauma (*Novák et al., 2016*; *Parameshwarappa et al., 2022*; *Schormans et al., 2019*). Individual L2/3 PyrNs exhibited strong tone-evoked transients and well-organized tonal receptive fields (*Figure 3D*) that were measured simultaneously across hundreds of PyrNs to reveal a coarse low- to high-frequency tonotopic gradient. We used a support vector machine (SVM) to bisect the pre-exposure A1 tonotopic map into a low-frequency intact zone and high-frequency deafferented zone (*Figure 3E*, top). By returning to the same A1 region for imaging every 1–3 days (*Figure 3F*), we confirmed that L2/3 PyrNs shifted their preferred frequency toward frequencies bordering the cochlear lesion (*Figure 3E*, bottom). Analysis of all tone-responsive PyrNs ($n$ = 1749 in four trauma mice; $n$ = 1748 in four sham mice) demonstrated that tonotopic remapping was limited to the deafferented zone (*Figure 3G*), where the percentage of L2/3 PyrNs preferentially tuned to lesion edge frequencies more than doubled following acoustic trauma (*Figure 3H*) without any systematic change in response threshold (*Figure 3I*).

A marked increase in PyrNs tuned to lesion edge frequencies could contribute to the enhanced perceptual sensitivity to 8 kHz tones identified in behavioral experiments (*Figure 2*). To address the potential association between the neural and perceptual changes more explicitly, we measured the growth of PyrN responses across sound intensities both as a function of where neurons were located relative to the deafferentation boundary and when – following noise exposure – the measurement was made (*Figure 4*). We first quantified neural response gain as the change in PyrN response with increasing sound intensity, where the particular range of intensities used for the calculation was determined from the overall shape of the growth function (*Figure 5A*, *Figure 5—figure supplement 1A*).

We noted that neural gain was broadly enhanced across the tonotopic map for the first several days following trauma but then receded to the high-frequency deafferented region in measurements made 1 week or more following noise exposure (*Figure 5B*). Next, we expanded the neural gain analysis in sham and trauma mice to four stimulus frequencies: a high-frequency tone aligned to the damaged cochlear region (32 kHz), a spared low-frequency tone far from the cochlear lesion (5.7 kHz), and two spared mid-frequency tones near the edge of the cochlear lesion (8 and 11.3 kHz). Neural gain at each of the four test frequencies was measured separately for intact and deafferented cortical zones and expressed in units of fold change relative to baseline gain measured from the corresponding population response prior to noise exposure. This analysis identified several clear results: (1) a strong initial uptick in neural gain measured in both topographic regions following trauma; (2) persistent (lasting greater than 1 week) increases in neural gain were observed only for spared mid-frequency tones in deafferented cortical regions; (3) no significant changes in neural gain were observed in sham-exposed mice (*Figure 5C*). Thus, excess central gain reflected the interaction of four factors: (1) *whether* the initial sound exposure induced SNHL, (2) *where* within the cortical frequency map the cell is located, (3) *when* relative to exposure the measurement is made, and (4) the proximity of the stimulus test frequency to the cochlear lesion.

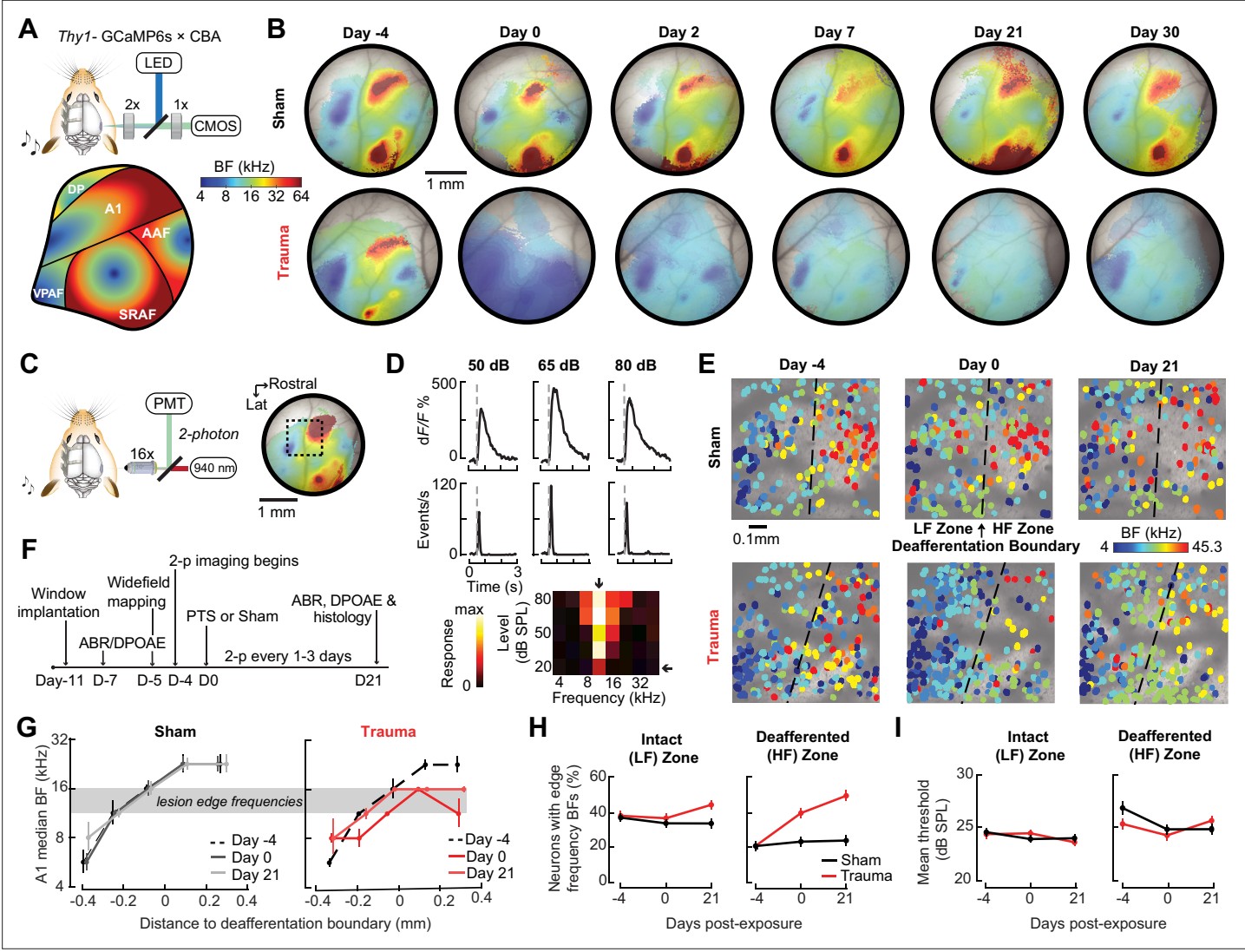

**Figure 3.** Tonotopic remapping within the cortical deafferentation zone revealed by chronic mesoscale and cellular calcium imaging. (**A**) *Top*, approach for widefield calcium imaging using a tandem lens epifluorescence microscope in awake Thy1-GCaMP6s × CBA/CaJ mice that express GCaMP6s in pyramidal neurons (PyrNs). *Bottom*, schematic depicts the typical arrangement of individual fields within the ACtx based on tonotopic best frequency (BF) gradients (as detailed in *Romero et al., 2019*). (**B**) Chronic widefield BF maps in example sham and trauma mice (top and bottom, respectively) show BF remapping within the deafferented high-frequency regions throughout the ACtx after acoustic trauma. (**C**) *Left*, approach for chronic two-photon calcium imaging of layer 2/3 PyrN's along the A1 tonotopic gradient. *Right*, two-photon imaging field-of-view superimposed on the widefield BF map measured in another example mouse. (**D**) *Top*, example of tone-evoked GCaMP transients measured as the fractional change in fluorescence and deconvolved activity. *Bottom*, peak deconvolved amplitudes for tones of varying frequencies and levels are used to populate the complete frequency-response area and derive the BF (downward arrow) and threshold (leftward arrow) for each neuron. (**E**) BF arrangements in L2/3 PyrNs measured at three times over the course of a month in representative sham and trauma mice. A support vector machine (SVM) was trained to bisect the low- and high-frequency zones of the A1 BF map (LF [<16 kHz] and HF [≥16 kHz], respectively). The dashed line represents the SVM-derived boundary to segregate the LF and HF regions. The SVM line is determined for each mouse on day −4 and then applied to the same physical location for all future imaging sessions following alignment. (**F**) Timeline for chronic two-photon imaging and cochlear function testing in each sham and trauma mouse. (**G**) Individual PyrNs are placed into five distance categories based on their Euclidean distance to the SVM line and the BF of each category is expressed as the median ± bootstrapped error. Following trauma, BFs in the HF zone are remapped to sound frequencies at the edge of the cochlear lesion. (**H**) Across all tone-responsive PyrNs measured at three time points, the percent of neurons with BFs corresponding to edge frequencies (11.3–16 kHz) was greater in trauma mice (N = 4 mice, n = 1749 PyrNs) than sham (N = 4 mice, n = 1748 PyrNs), was greater in the deafferented HF region than the intact LF region, and increased over time in trauma mice compared to sham controls (three-way analysis of variance [ANOVA] with Group, Region, and Time as factors: main effect for Group, $F = 34.29$, $p = 5 × 10^{-9}$; Group × Region interaction term, $F = 7.42$, $p = 0.007$; Group × Time interaction term, $F = 10.17$, $p = 0.00004$). (**I**) Competitive expansion of edge frequency BFs in the deafferented HF zone was not accompanied by a change in neural response threshold (three-way ANOVA: main effect for Group, $F = 0.8$, $p = 0.37$; Group × Region interaction term, $F = 0.93$, $p = 0.33$; Group × Time interaction term, $F = 1.33$, $p = 0.27$).

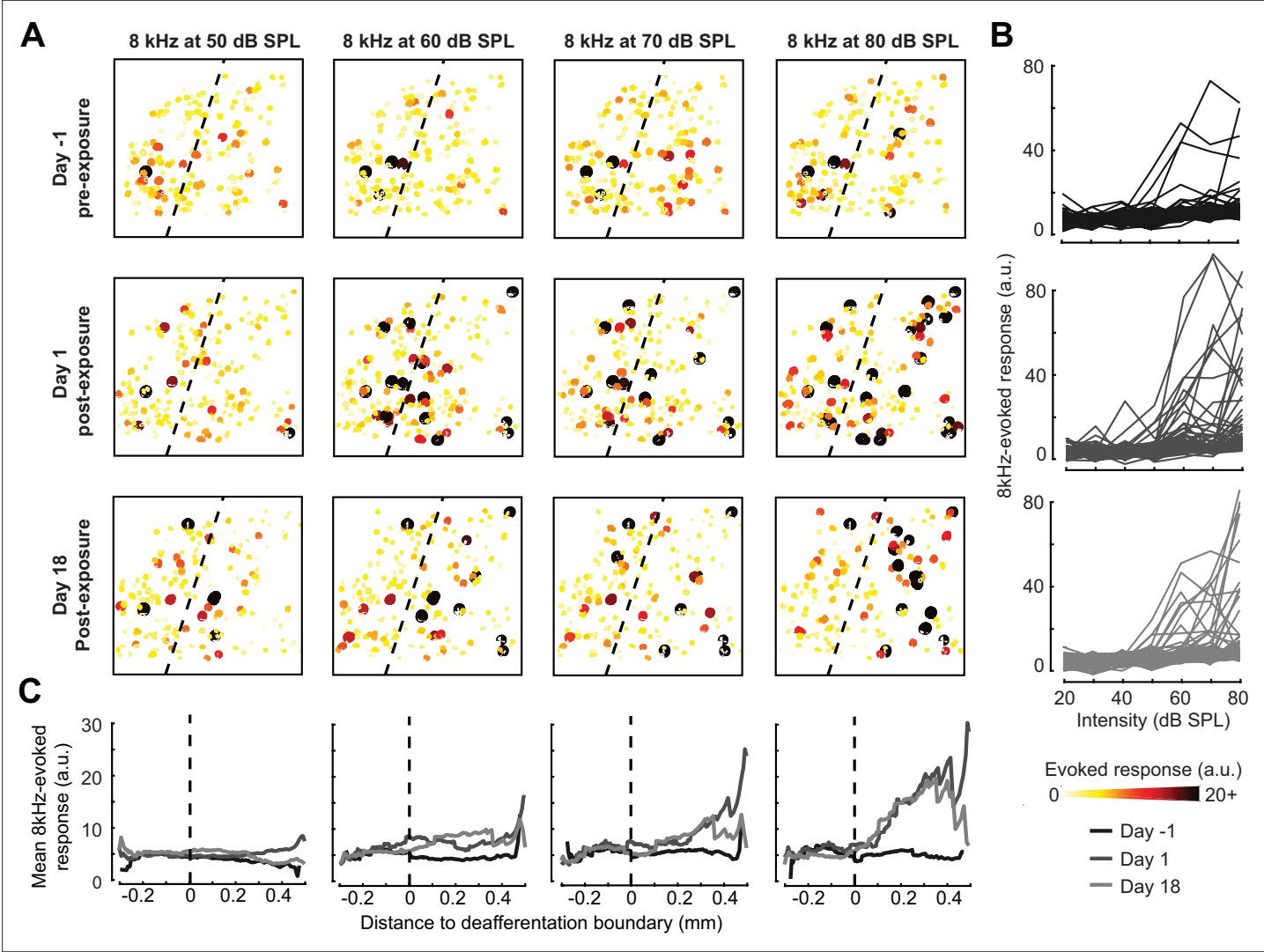

**Figure 4.** Spatial and temporal expression of excess central gain following acoustic trauma. (**A**) Response amplitudes to 8 kHz tones presented at varying levels (columns) and days (rows) relative to noise exposure for 185 PyrNs in an example mouse. (**B**) Intensity-response functions for 66 randomly selected PyrNs recorded on days −4, 1, and 18 relative to the day of noise exposure. (**C**) Mean tone-evoked responses for all PyrNs relative to the support vector machine (SVM) deafferentation boundary at 50–80 dB SPL plotted separately for each of the 3 days.

## Cortical hyperresponsivity and increased gain mirrors behavioral hypersensitivity to spared mid-frequency inputs

In the Go/NoGo tone detection behavior, mice exhibited steepened 8 kHz detection functions after trauma (*Figure 2B*). Excess gain in A1 L2/3 PyrNs mirrored this result in that steepened neural growth functions were observed for 8 kHz tones over the same timescale. To more directly relate ACtx hyper-activity to perceptual hypersensitivity, we used a decoder to categorize the presence or absence of sound based on single-trial responses from hundreds of simultaneously recorded neurons located either in the intact or deafferented zone of the A1 map. This was accomplished by training an SVM classifier on PCA-decomposed population activity during short periods of tone presentation or silence (*Figure 5D*). Classification of single trial A1 ensemble responses supported the hypothesis that cortical discrimination of sound versus silence would be enhanced for low- and mid-intensity 8 kHz tones but reduced for 32 kHz tones after trauma (*Figure 5E*). Enhanced neural detection of 8 kHz tones was largely driven by PyrNs in deafferented map regions, whereas the loss of cortical sensitivity to high-frequency tones was observed in both topographic zones (*Figure 5F*).

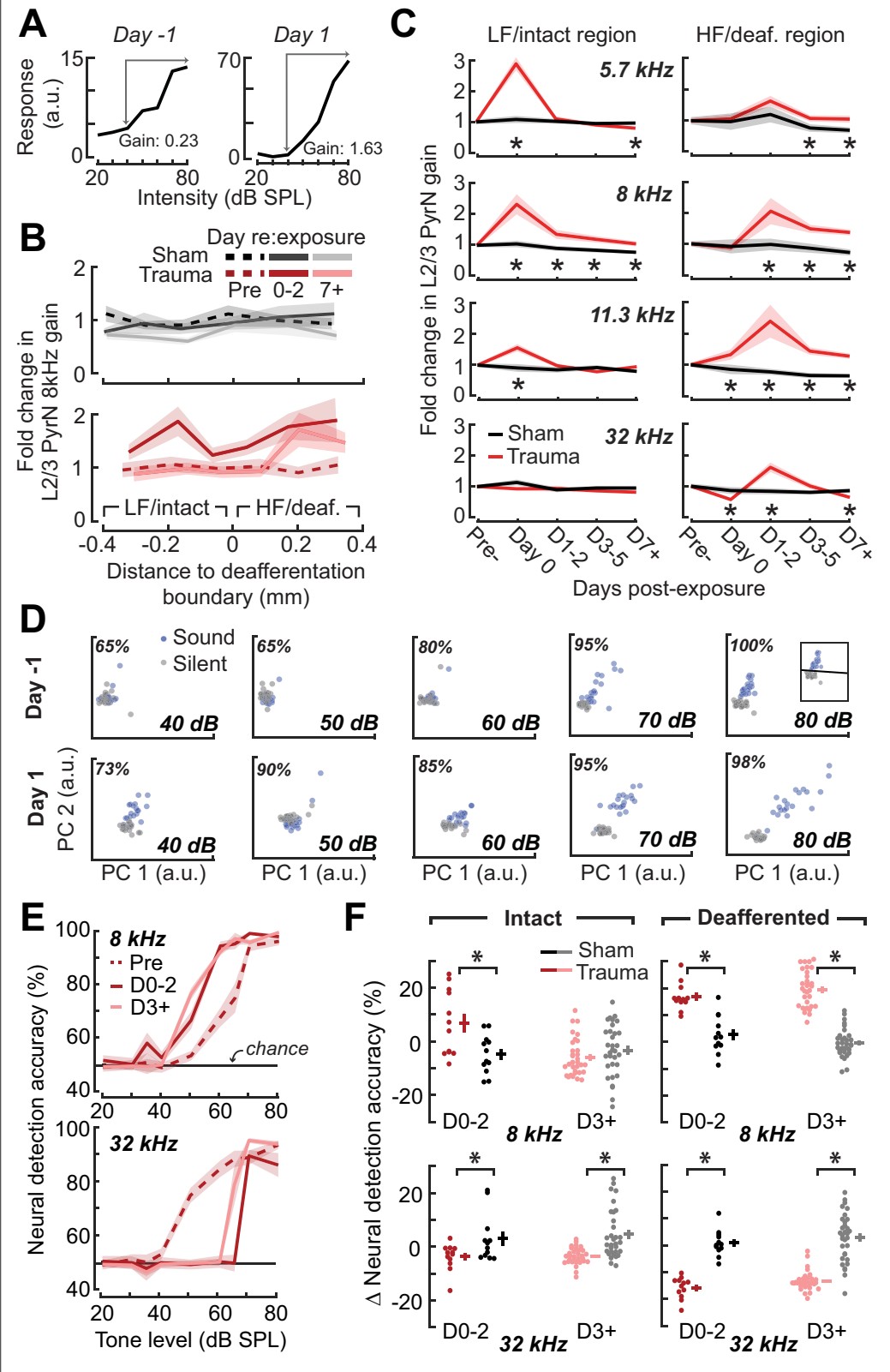

**Figure 5.** Increased central gain is associated with hypersensitive neural encoding of low-intensity sounds. (**A**) Neural gain is measured as the average rate of response growth in the sound level-response function. A detailed description of how neural gain is measured for different types of level-response functions is provided in *Figure 5— figure supplement 1*. (**B**) Mean change in response gain to an 8 kHz tone relative to the baseline support vector

*Figure 5 continued on next page*

*Figure 5 continued*

machine (SVM) demarcation of the low-frequency (LF) intact and high-frequency (HF) deafferented regions of the A1 map from all sham-exposed (*top*) and all noise-exposed (*bottom*) mice. (**C**) Fold change in 8 kHz response gain relative to the pre-exposure period in sham (n = 23,007 PyrNs from four mice) and trauma (n = 23,319 PyrNs from four mice). After acoustic trauma, the response gain for low- and mid-frequency tones is temporarily increased in the intact region. A sustained increase in response gain is observed in the deafferented region, particularly for tone frequencies bordering the cochlear lesion. Four-way analysis of variance (ANOVA) with Group, Region, Time, and Frequency as factors main effects, respectively: $F = 111.03$, $p = 6 \times 10^{-26}$; $F = 0.03$, $p = 0.87$; $F = 23.21$, $p = 4 \times 10^{-19}$; $F = 9.87$, $p = 2 \times 10^{-6}$; Group × Region × Time × Frequency interaction term: $F = 2.23$; $p = 0.008$. Asterisks denote significant pairwise post hoc differences between groups ($p < 0.05$). (**D**) Neural ensemble responses to single trials of sound or silence were decomposed into principal components (PC) and classified with an SVM decoder. The first two PCs are presented from an example mouse 1 day before or after acoustic trauma for an 8 kHz tone. Single trial classification accuracy is provided for each sound intensity. (**E**) Mean decoding accuracy for 8 and 32 kHz tones across all noise-exposed mice as a function of sound intensity at varying times following acoustic trauma. (**F**) Mean change in decoding accuracy across all intensities for 8 and 32 kHz tones for L2/3 PyrNs in the intact and deafferented regions of the A1 map. For 8 kHz tones, PyrN ensemble decoding shows sustained improvement in the deafferented region but a temporary improvement in the intact region. Ensemble decoding of 32 kHz tones is reduced for all time points and measurement regions. Dots represent single imaging sessions. Bars denote mean ± standard error of the mean (SEM). Asterisks represent significant differences with unpaired *t*-tests ($p < 0.05$).

The online version of this article includes the following figure supplement(s) for figure 5:

**Figure supplement 1.** Assessment of gain measurement for different intensity-response function shapes.

## Tracking changes in activity and local synchrony in individual PyrNs over several weeks

A principal advantage of chronic two-photon calcium imaging lies in the ability to perform longitudinal assessments of activity changes in individual cells over relatively long time periods, essentially enabling individual cells to serve as their own control when evaluating changes after noise exposure. To track individual PyrNs over a several week period, imaging fields were first registered to a pre-exposure imaging session, and cell tracking was performed using probabilistic modeling (*Sheintuch et al., 2017*; *Figure 6A, B*, see Materials and methods). Some PyrNs could be tracked throughout all 15 imaging sessions and others just for a single session, where the overall number of neurons depended on how liberal or conservative the threshold was set for chronic tracking confidence (*Figure 6—figure supplement 1B*). We set the criterion for identifying chronically tracked cells by creating a control scenario in which a single imaging field from eight different mice were concatenated, allowing us to quantify the occurrence of falsely identified tracked cells across sessions. We observed a clear separation in the occurrence of veridically and falsely identified cells across sessions, where falsely tracked PyrNs over eight or more sessions with a confidence threshold of 0.8 were never observed, prompting us to use this criterion for the remainder of our analyses.

Interestingly, we noted that new active PyrNs appeared immediately following noise exposure, while PyrNs tracked throughout the baseline imaging sessions disappeared (*Figure 6—figure supplement 1C*). Because the degree of turnover was far less after sham exposure, because the appearance or disappearance of PyrNs after acoustic trauma was concentrated near the deafferentation boundary (*Figure 6—figure supplement 1D*), and because approximately 75% of PyrN disappearance occurred within the 48 hr after acoustic trauma (*Figure 6—figure supplement 1E*), we concluded that increased PyrN turnover in the imaging field must be related to cortical changes arising indirectly from the cochlear SNHL. Possibilities include large and heterogenous state shifts in activity (from virtually quiescent to active or vice versa) as well as changes in the physical topology of the cortex due to degradation of extracellular matrix proteins and increased neurite motility after sudden hearing loss (*Nguyen et al., 2017*; *Tschida and Mooney, 2012*).

Chronic cell tracking allowed us to revisit analyses of central gain at the level of individual PyrNs, rather than across PyrN populations in intact or deafferented map regions. At this higher level of spatial specificity, we noted PyrNs exhibiting no increase or a temporary increase in response to an 8 kHz tone from cells in the intact zone and a permanent increase in responsiveness from cells in the deafferented zone (*Figure 6C*). More specifically, we noted transient hyper-responsiveness from 20 to 80 dB SPL for all neurons across A1 (*Figure 6D*) followed by a second stage of 8 kHz hyper-responsivity

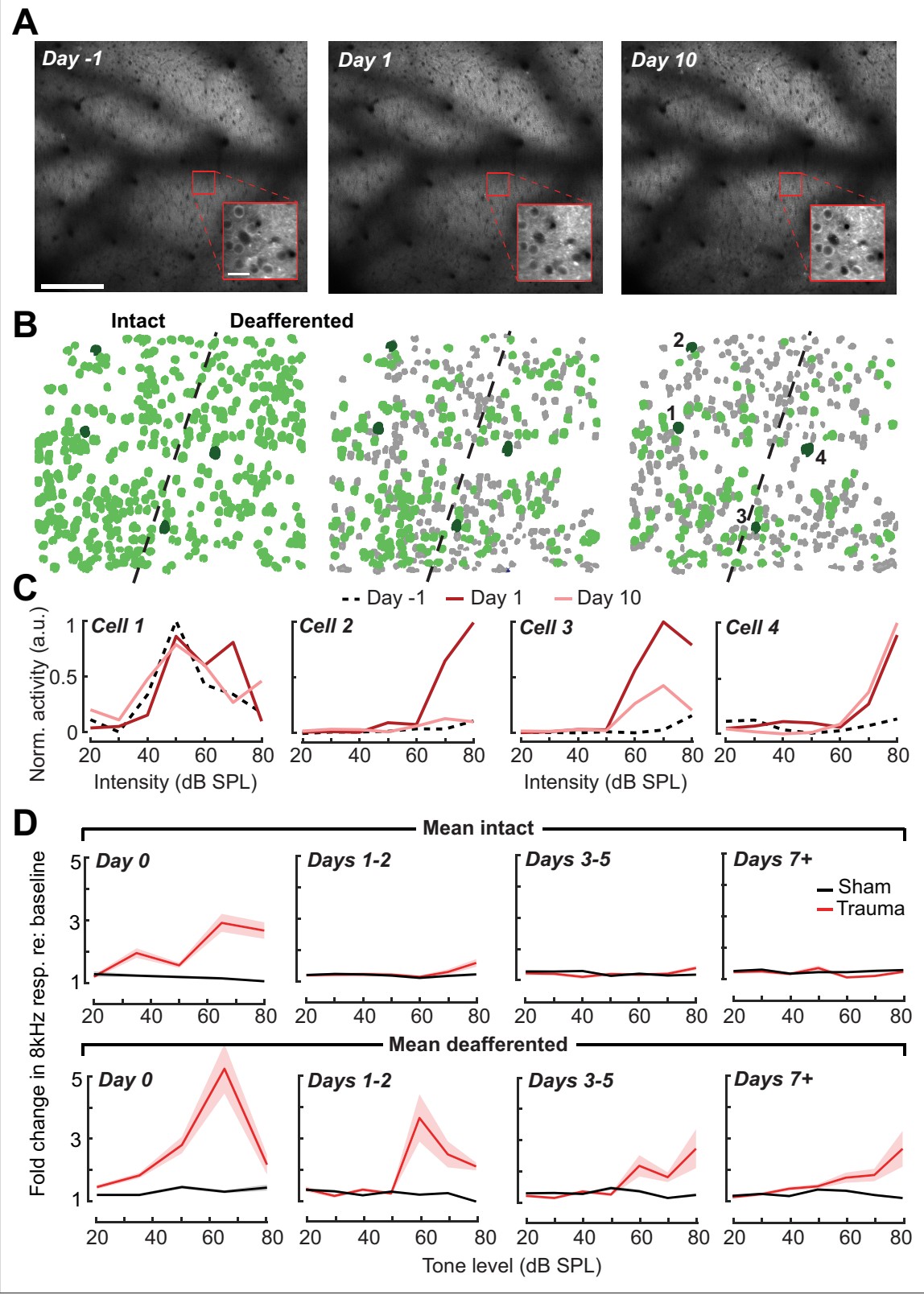

**Figure 6.** Tracking single neuron response gain dynamics over a several week period before and after noise exposure. (**A**) Example fields-of-view from a single mouse showing the same imaging field over a several week period. Insets show data acquired at ×4 digital zoom. Scale bar is 200 μm, inset scale bar is 20 μm. (**B**) Single L2/3 PyrN ROI masks. Green masks indicate cells found on the current day and all previous days using a cell score threshold of 0.8 (see *Figure 6—figure supplement 1*). (**C**) Normalized 8 kHz intensity-response functions for the four PyrNs highlighted in *B*. Neurons in the intact

*Figure 6 continued on next page*

*Figure 6 continued*

region show temporary increases in their responses while neurons in the deafferented region show permanent hyperresponsiveness. (**D**) Mean fold change in response to 8 kHz tones of varying intensities for individual neurons relative to their own response function measured prior to noise exposure ($n$ = 303/552, trauma/sham). Gain is strongly elevated in both regions hours after trauma. Sustained gain increases are observed in the deafferented zone for at least 1 week following trauma but not in the intact zone. Four-way analysis of variance (ANOVA) with Group and Region as factors, and Time and Intensity as repeated measures (main effects, respectively: $F$ = 6.87, p = 0.01; $F$ = 2.9, p = 0.09; $F$ = 8.69, p = 4 × 10$^{-5}$; $F$ = 116.61, p = 8 × 10$^{-13}$; Group × Region × Time × Intensity interaction term: $F$ = 6.65; p = 0.0004).

The online version of this article includes the following figure supplement(s) for figure 6:

**Figure supplement 1.** Validation of single cell tracking over imaging days.

restricted to intensities above 50 dB SPL in the deafferented zone (***Figure 6D***). These observations can account for performance changes in the population-level decoder (***Figure 5E***) and more generally confirm that the temporal and spatial patterns noted across populations of PyrNs are supported by commensurate plasticity of single PyrN responses.

Prior studies have noted increased spontaneous activity in acute single unit or multiunit recordings in the days following acoustic trauma, though it has not been clear whether that is driven by the unmasking of hitherto silent neurons (***Figure 6—figure supplement 1C***) or increased activity rates of individual neurons over time (***Kotak et al., 2005***; ***Noreña et al., 2010***; ***Noreña et al., 2003***; ***Seki and Eggermont, 2003***). Normalizing by the pre-exposure period, we noted an approximate 20% increase in PyrN spontaneous activity across A1 after acoustic trauma, but relatively modest effects following sham exposure (***Figure 7A, B***). Unlike the increase in edge frequency tone-responsiveness, increased spontaneous activity after acoustic trauma was not topographically restricted and continued to increase steadily with time after acoustic trauma (***Figure 7B, C***).

Tinnitus and auditory hypersensitivity are thought to arise from a combination of spontaneous hyperactivity as well as increased synchrony between cells (***Auerbach et al., 2019***, ***Auerbach et al., 2014***; ***Resnik and Polley, 2021***; ***Seki and Eggermont, 2003***; ***Shore and Wu, 2019***). To determine how increased synchrony developed over topographic space and post-trauma time, we cross-correlated periods of spontaneous activity between pairs of A1 PyrNs and removed the influence of gross changes in activity rates through shuffle correction (***Figure 7D***, see Materials and methods). Using the area under the shuffle-corrected cross-correlogram as our index of pairwise synchrony (***Figure 7D***), we noted that synchronized activity normally decayed to asymptote once PyrNs were separated by more than 0.25 mm (***Figure 7E***, top). In the first 72 hr following acoustic trauma, pairwise synchrony was significantly enhanced for PyrNs separated by as much as 0.7 mm (***Figure 7E***, bottom). Three days after trauma and beyond, local synchrony remained strongly elevated for PyrNs relative both to baseline and to sham-exposed controls. Increased synchrony following trauma was primarily observed in PyrN's located close to – or straddling – the deafferentation boundary, where the peak of elevated synchrony was positioned within a tonotopic region corresponding to lesion edge sound frequencies (***Figure 7F***).

## Predicting the degree of excess central gain after trauma in individual PyrNs based on their pre-exposure response features

Taken together, these observations also suggest that plasticity following acoustic trauma is organized into two phases: a dynamic phase during the first 48–72 hr after noise exposure involving topographically widespread hyper-correlation, hyper-responsivity to mid-frequency sounds in the intact map regions, and large-scale turnover in PyrN stability around the deafferentation map boundary, followed by a stable phase beginning 3 days after exposure where gain is increased for spared tone frequencies in deafferented map regions. However, even when tracking the same neuron over time during the stable phase of reorganization, we still noted considerable heterogeneity in the expression of central gain changes between individual PyrNs. For example, for two PyrNs located within the deafferented map region, one could show stable 8 kHz growth functions after trauma (***Figure 8A***) and the other excess gain (***Figure 8B***).

To account for unexplained variability in central gain changes, we asked whether response features measured in the pre-exposure baseline period could predict whether neurons would express homeostatic or non-homeostatic regulation of sound-evoked activity. Returning to the same two example neurons, we noted that the PyrN that maintained stable gain had a relatively low spontaneous activity

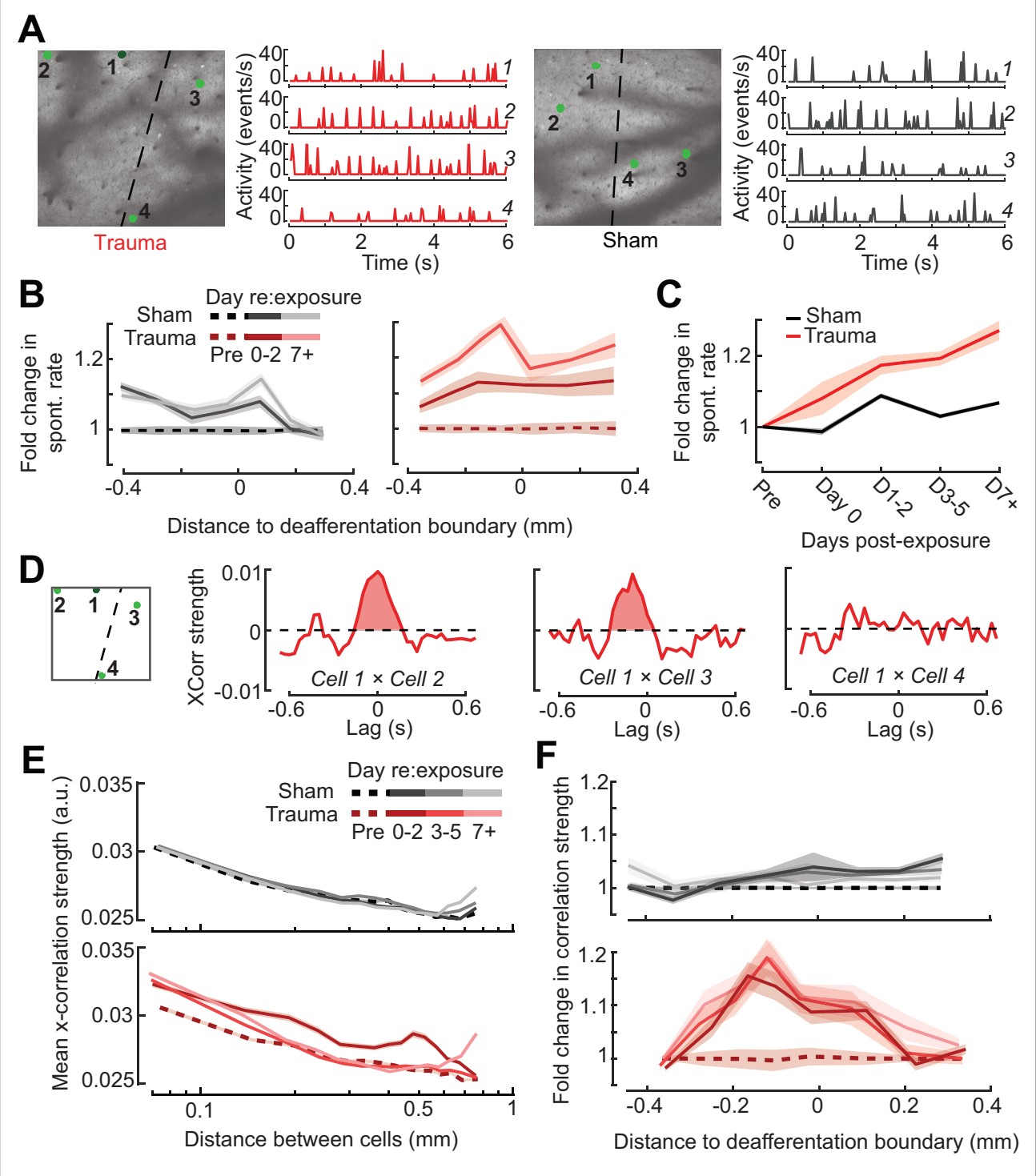

**Figure 7.** Topographic regulation of neural hyperexcitability and hyper-synchrony after acoustic trauma. (**A**) Spontaneous activity traces in four example neurons from a trauma (*left*) and sham (*right*) mouse. (**B**) In chronically tracked PyrNs, spontaneous activity changes are expressed as fold change relative to that cell's pre-exposure baseline. Increased spontaneous activity after trauma (*right*) or the lack thereof after sham exposure (*left*) are plotted over topographic distance and over post-exposure time. (**C**) Spontaneous activity changes across the cortical map are significantly greater after trauma than sham exposure and increase over post-exposure time (n = 915/1125 tracked cells, for trauma/sham; mixed model analysis of variance (ANOVA) with Group as a factor and Time as a repeated measure, main effect for Group [F = 12.81, p = 0.0004], main effect for Time [F = 65.03, p = 3 × 10⁻⁴⁰], Group × Time interaction term [F = 17.66, P = 3 × 10⁻¹¹]). (**D**) Synchrony in the spontaneous activity of PyrN pairs is measured as the area under the shuffle-corrected cross-correlogram peak (shaded red region, see Materials and methods). Example data are plotted for the same four PyrNs with topographic

*Figure 7 continued on next page*

*Figure 7 continued*

positions indicated in left panel. (**E**) Looking across all significantly correlated PyrN pairs recorded in a given imaging session ($n$ = 3,301,363 pairs, 1,624,195/1,677,168 for trauma/sham), neural synchrony is reduced as the physical separation between somatic ROIs increases. Synchrony is increased after trauma, though remains elevated only among nearby PyrNs (three-way ANOVA with Group, Day, and Distance as factors: main effects for Group [$F$ = 556.94, p = 4 × 10$^{-123}$], Day [$F$ = 82.6, p = 2 × 10$^{-53}$], and Distance [$F$ = 8527.73, p = 0], Group × Day × Distance interaction term [$F$ = 7.94, p = 3 × 10$^{-5}$]). (**F**) For each chronically tracked neuron (same sample as *C*), we calculate their average neural synchrony with all other cells (only taking significant pairs). Given the location of these tracked cells, we can examine the fold change in neural synchrony relative to pre-exposure baseline across the topographic map. Neural synchrony is significantly and stably increased after trauma, particularly for PyrNs located near the deafferentation boundary (mixed model ANOVA with Group and Distance as factors and Day as a repeated measure: main effects for Group [$F$ = 26.62, p = 3 × 10$^{-7}$], Day [$F$ = 1.68, p = 0.19], and Distance [$F$ = 0.53, p = 0.47], Group × Distance interaction term [$F$ = 5.53, p = 0.02]).

rate and sharp frequency tuning prior to acoustic trauma, whereas the PyrN that would express excess gain had higher spontaneous activity and a relatively broad pure tone receptive field (*Figure 8B, C*). Expanding this analysis to additional example neurons over post-exposure time further suggests that PyrNs that would go on to express non-homeostatic auditory growth functions tended to feature higher spontaneous activity levels at baseline (*Figure 8D*).

We used a multiple linear regression model to better capture how baseline properties can explain heterogenous changes in neural response gain. Central gain was operationally defined as the change in the area under the 8 kHz growth function relative to the PyrN's baseline growth function area (as per *Figure 8A, B*). Quantifying neural gain as fold change in the normalized 8 kHz growth function area recapitulated the same topographic and temporal dependence described above with intensity growth slope (*Figure 8—figure supplement 1A*). To avoid overfitting the regression model, we selected a single time point – 3–5 days after noise exposure – that corresponded to the stable phase of reorganization after acoustic trauma, while maximizing our sample size of chronically tracked PyrNs. As predictor variables, we selected various spontaneous and sound-evoked response baseline features along with features related to the physical position of the PyrN within the A1 tonotopic map (see *Figure 8* legend).

We found that regressing the post-exposure change in neural gain on baseline PyrN's response features could account for approximately 40% of the variance in central gain changes after acoustic trauma but just 13% in sham-exposed controls, where neural gain changes were small overall and far less systematic (*Figure 8E*). This is noteworthy because the model excluded features related to the degree of cochlear damage or reduced bottom-up sensory afferent drive, which are traditionally interpreted as the primary determinants of cortical central gain changes. To determine the weighting of each predictor variable to the overall model fit, we randomly shuffled each variable and refit the model to calculate the decrement in the adjusted $R$-squared. The results are provided for all univariate predictors (*Figure 8F*) and all predictors including interaction terms (*Figure 8—figure supplement 1B*). We observed that excess non-homeostatic gain regulation following acoustic trauma occurred with the highest probability in PyrNs exhibiting weak, monotonically increasing 8 kHz growth functions and higher spontaneous activity levels at baseline (*Figure 8G*). Further, PyrNs located in the high-frequency (deafferented) region but with a lower frequency BF further increased the likelihood of expressing excess neural gain after trauma (*Figure 8G*). Taken together, these findings show that certain idiosyncratic response features measured just prior to acoustic trauma can predict whether A1 L2/3 PyrNs will undergo stable, homeostatic regulation or excess, non-homeostatic changes to a stimulus positioned near the edge of the cochlear lesion.

## Discussion

Here, we introduced a noise exposure protocol that damages OHCs and neural afferents in the high-frequency base of the cochlea, providing a mouse model for the common steeply sloping high-frequency hearing loss profile in humans that is often associated with tinnitus, loudness hypersensitivity, and poor multitalker speech intelligibility (*Figure 1*). We demonstrated that behavioral hypersensitivity to spared, mid-frequency tones was also observed for direct stimulation of thalamocortical projection neurons, identifying the ACtx as a potential locus of plasticity underlying auditory hypersensitivity (*Figure 2*). We tracked ensembles of excitatory PyrNs over several weeks and confirmed large-scale reorganization of ACtx tonotopic maps (*Figure 3*) and sound intensity coding (*Figure 4*) that recapitulated the auditory hypersensitivity documented behaviorally (*Figure 5*). Neural

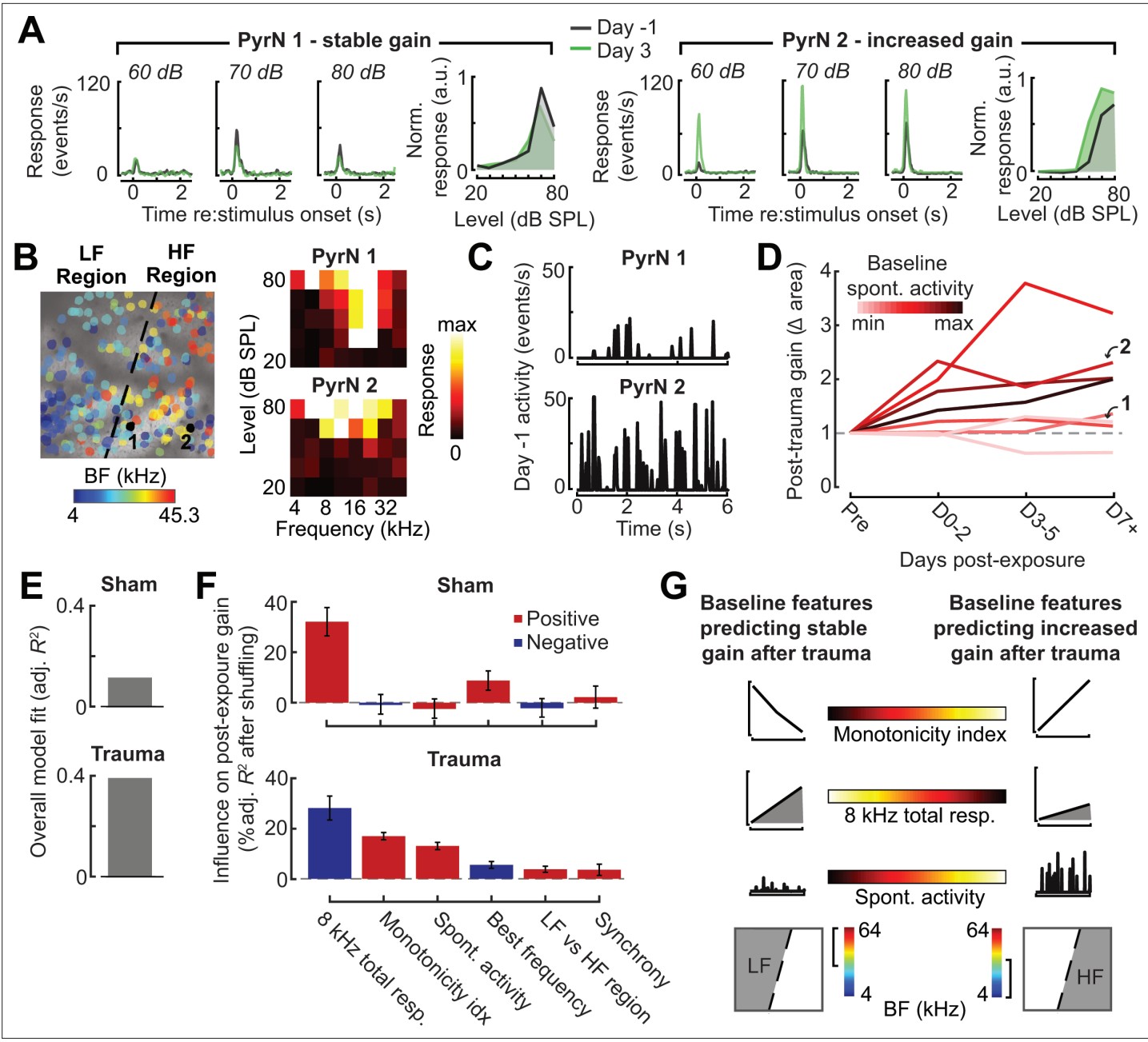

**Figure 8.** Identifying baseline features in single PyrNs that predict stable versus excess gain changes after acoustic trauma. (**A**) Two exemplar tracked neurons illustrating stable (*left*) and excess (*right*) response growth to an 8 kHz tone following acoustic trauma. (**B**) Both neurons are located in the deafferented map region but had different best frequencies (BFs) and frequency tuning properties measured during the baseline imaging session. (**C**) Spontaneous activity for the same two PyrNs also differed at baseline. (**D**) For tracked neurons, gain is measured as the fold change in the area under the intensity-response growth function relative to the pre-exposure baseline (see *Figure 8—figure supplement 1A*). In eight representative neurons, a higher spontaneous activity rate at baseline was associated with excess central gain after trauma. Arrows denote PyrNs 1 and 2 shown in *A–C*. (**E**) A linear model used various pre-exposure properties of chronically tracked neurons to predict their change in gain (see Materials and methods). Models were fit separately for PyrNs recorded from trauma (*n* = 510 cells) and sham (*n* = 749 cells) mice. The response variable is defined as the area under the 8 kHz response curve after noise/sham exposure relative to this same area measurement in baseline. (**F**) For each model, individual predictor variables were shuffled and the models were refit. The resulting decrease in the adjusted *R*-squared is shown for variables in both models, and bars are color coded by the sign of the relationship of each predictor variable with the response variable. Errors are bootstrapped. For the full model see *Figure 8—figure supplement 1B*. Predictor variables in order: area under the baseline 8 kHz intensity-response growth function, monotonicity index for the 8 kHz intensity-response function defined as the response at the maximum intensity divided by the response at the best intensity, mean spontaneous activity, BF, an indicator variable for whether the cell is in the deafferented or intact region, and the strength of correlated activity between the PyrN and its

*Figure 8 continued on next page*

*Figure 8 continued*

neighbors. (**G**) A graphical summary of the linear model results schematize the baseline factors most strongly associated with stable (*left*) or excess (*right*) gain after trauma.

The online version of this article includes the following figure supplement(s) for figure 8:

**Figure supplement 1.** Extended description of the multivariate linear model analysis in sham and trauma mice.

hyperresponsivity to spared mid-frequency sounds was accompanied by hyperactive and hyper-correlated spontaneous activity (*Figure 7*), where the degree of excess neural gain following acoustic trauma in individual neurons could be predicted from many of these response features measured during the pre-exposure baseline period (*Figure 8*). Collectively, these findings underscore the close association between excess cortical gain and disordered sound perception after cochlear sensori-neural damage and identify activity signatures that predispose neurons to non-homeostatic hyperac-tivity following noise-induced hearing loss.

## Underlying mechanisms

Auditory hypersensitivity is a common auditory perceptual complaint associated with SNHL. Excess central gain – an abnormally steep growth of neural response with sound intensity – is the hypoth-esized neural substrate of auditory hypersensitivity (*Auerbach et al., 2019*, *Auerbach et al., 2014*; *Zeng, 2020*; *Zeng, 2013*). Excess central gain is prominently expressed in the ACtx of animals with sensorineural cochlear damage (*Asokan et al., 2018*; *Chambers et al., 2016a*; *Noreña et al., 2003*; *Parameshwarappa et al., 2022*; *Popelár et al., 1987*; *Qiu et al., 2000*; *Resnik and Polley, 2017*, *Resnik and Polley, 2021*; *Seki and Eggermont, 2003*; *Syka et al., 1994*) by contrast to the auditory nerve, where sound-evoked neural responses are strongly reduced (*Heinz et al., 2005*; *Heinz and Young, 2004*; *Wake et al., 1993*). Reorganization is also observed in subcortical stations of sound processing in animals with sensorineural cochlear damage (*Chambers et al., 2016b*; *Kamke et al., 2003*; *Niu et al., 2013*; *Schrode et al., 2018*; *Shaheen and Liberman, 2018*), but only particular cell types (*Cai et al., 2009*) and – in studies that have made direct inter-regional comparisons – is less robust overall than neural gain changes at the level of ACtx (*Chambers et al., 2016a*; *Qiu et al., 2000*).

Our findings build upon this literature by demonstrating excess central gain in L2/3 excitatory PyrNs that resembled behavioral hypersensitivity to spared mid-frequency tones. Seminal work demonstrated that animals became behaviorally hypersensitive to electrical microstimulation of central auditory neurons following cochlear damage (*Gerken, 1979*). Here, we expanded on this observation by demonstrating that noise-exposed mice are behaviorally hypersensitive to direct activation of thalamocortical projection neurons, which further underscores the association between excess cortical gain and auditory hypersensitivity. Hypersensitivity to direct stimulation of the thalam-ocortical pathway may be underpinned by changes in ACtx gene expression that result in elevated mRNA levels for glutamate receptor subunits and reduced mRNA transcripts and membrane-bound protein expression for GABA$_A$ receptor units (*Balaram et al., 2019*; *Sarro et al., 2008*). These adjust-ments in ligand-gated receptors have been associated with commensurate elevations in spontaneous excitatory postsynaptic currents and reduced inhibitory postsynaptic currents in PyrNs (*Kotak et al., 2005*; *Yang et al., 2011*). Thus, disinhibition and hypersensitization are conceptualized as synergistic compensatory responses that are triggered by a sudden decline in peripheral neural input, rendering cortical PyrNs less sensitive to feedforward inhibition from local inhibitory neurons – from parvalbumin-expressing (PV) fast-spiking interneurons in particular – and hyperresponsive to sound (*Masri et al., 2021*; *Resnik and Polley, 2021*; *Resnik and Polley, 2017*).

One challenge to this purely bottom-up model for compensatory plasticity underlying excess central gain is that it cannot readily account for why neighboring neurons exhibit heterogenous changes in sound-evoked hyperresponsivity after peripheral damage. Here, we found that approxi-mately 40% of the variability in excess central gain among individual PyrNs could be accounted for by their response properties measured in the days just prior to acoustic trauma. In particular, PyrNs with low spontaneous activity rates and non-monotonic encoding of sound intensity – features associated with stronger intracortical inhibition (*Tan et al., 2007*; *Wu et al., 2006*) – showed more stable gain control after trauma. Conversely, initially high spontaneous activity, hyper-correlation, and gradual

monotonic intensity growth functions could be reflective of weaker intracortical inhibitory tone and were features that predicted non-homeostatic excess gain after acoustic trauma.

Several recent findings support the idea that variations in the strength of intracortical inhibition can function as a watershed for subsequent functional outcomes. In an earlier study, we applied ouabain to the cochlear round window to produce bilateral lesions of approximately 95% of cochlear afferent neurons. Near-complete elimination of cochlear afferent input was associated with functional deafness at the level of the ACtx in approximately half the animals but a remarkable recovery of sound responsiveness in the other half. Single unit recordings from the cohort of mice that recovered sound processing weeks after cochlear neural loss all featured a rapid decline in PV-mediated feedforward inhibition onto PyrNs during the first hours and days following the peripheral injury, while the mice that showed no functional recovery expressed a far slower decay in PV-mediated inhibition (*Resnik and Polley, 2017*). Another piece of evidence comes from ACtx single unit recordings in marmosets outfitted with unilateral cochlear implants. Single units with narrowly tuned non-monotonic acoustic frequency-response areas – presumably reflecting stronger local inhibition – were suppressed by spatially diffuse electrical stimulation of the auditory nerve. Conversely, units with broad, V-shaped acoustic selectivity encoded cochlear implant stimulation with high fidelity (*Johnson et al., 2016*). Collectively, these findings underscore that the effects of peripheral afferent perturbations on central circuits are not exclusively determined by bottom-up drive but instead are also brokered by variations in the balance of excitation and inhibition within the local circuit or by gene expression differences between individual cells. At a global level, these mitigating influences are shaped by developmental stage (*Dorrn et al., 2010*; *Harris et al., 2005*; *Sun et al., 2010*) and circadian programs (*Basinou et al., 2017*), but – at the level of individual neurons – may reflect latent differences in genetic subtypes of L2/3 PyrNs or purely stochastic variations in inhibitory tone between different microcircuit milieus.

## Behavioral hypersensitivity: interpretations and technical limitations

We found evidence of behavioral hypersensitivity to sound following acoustic trauma as measured by a steeper relationship between increasing tone intensity and detection ability in a Go/NoGo task (*Figure 2*). Although previous indicators of behavioral hypersensitivity in animals have relied on more reflexive measures, such as the acoustic startle response (*Hickox and Liberman, 2014*; *Rybalko et al., 2015*; *Rybalko et al., 2011*; *Sun et al., 2012*), more recent work has utilized operant detection tasks including reaction times as an approximation of sound hypersensitivity (*Auerbach et al., 2019*). Here, we used an adaptive tracking approach to characterize the perceptual salience of liminal sound intensities spanning an approximate 30 dB range around threshold. To control for changes in global behavioral state due to acoustic trauma, our slope measurements are taken from functions using a sensitivity index (*d'*), which normalizes lick probability according to the false alarm rate determined from the delivery of catch (silent) trials. Thus, overall changes in stress, arousal, or other global behavioral states following acoustic trauma that impacted overall responsivity in catch and stimulus trials would be controlled for by the *d'* measurement. However, as with reaction time, the *d'* growth function is not a direct measure of loudness, per se, but instead is probably best likened to the change in stimulus salience for tones of varying intensity. Further, while the perceptual salience (*Figure 2*) and neural decoding of spared, 8 kHz tones (*Figure 5*) were both enhanced after high-frequency SNHL, these measurements were not performed in the same animals (and therefore not at the same time). Definitive proof that increased cortical gain is the neural substrate for auditory hypersensitivity after hearing loss would require concurrent monitoring and manipulations of cortical activity, which would be an important goal for future experiments.

While the findings presented here support an association between sensorineural peripheral injury, excess cortical gain, and behavioral hypersensitivity, they should not be interpreted as providing strong evidence for these factors in clinical conditions such as tinnitus or hyperacusis. Our data have nothing to say about tinnitus one way or the other, simply because we never studied a behavior that would indicate phantom sound perception. If anything, one might expect that mice experiencing a chronic phantom sound corresponding in frequency to the region of steeply sloping hearing loss would instead exhibit an increase in false alarms on high-frequency detection blocks after acoustic trauma, but this was not something we observed. Hyperacusis describes a spectrum of aversive auditory qualities including increased perceived loudness of moderate intensity sounds, a decrease in loudness tolerance, discomfort, pain, and even fear of sounds (*Pienkowski, 2014a*). The affective

components of hyperacusis are more challenging to index in animals, particularly using head-fixed behaviors, though progress is being made with active avoidance paradigms in freely moving animals (*Manohar et al., 2017*). Our noise-induced high-frequency SNHL and Go-NoGo operant detection behavior were not designed to model hyperacusis. Hearing loss is not strongly associated with hyperacusis, where many individuals have normal hearing or have a pattern of mild hearing loss that does not correspond to the frequency dependence of their auditory sensitivity (*Sheldrake et al., 2015*). While the excess central gain and behavioral hypersensitivity we describe here may be related to the sensory component of hyperacusis, this connection is tentative because it was elicited by acoustic trauma and because the detection behavior provides a measure of stimulus salience, but not the perceptual quality of loudness, per se.

### Cortical hyperreactivity: interpretations and technical limitations

Two-photon calcium imaging offers several key advantages for cortical plasticity studies including the ability to track single neurons over weeks (*Figure 6*) and genetic access to multiple cell types (*Resnik and Polley, 2021*). On the other hand, it can provide less insight into the mechanisms underlying destabilized excitatory-inhibitory balance than electrophysiological approaches (*Resnik and Polley, 2017*; *Yang et al., 2011*; *Yang et al., 2012*). Further, calcium indicators provide only an approximation of neural activity and can be limited in their kinetics and reliability to report precise cellular events (*Grienberger and Konnerth, 2012*), although correct deconvolution and post hoc analysis techniques can help to minimize issues introduced from calcium imaging (*Sabatini, 2019*).

Homeostatic Plasticity describes a negative feedback process that stabilizes neural activity levels following input perturbations. Homeostatic Plasticity mechanisms modify excitatory and inhibitory synapses over a period of hours or days to offset input perturbations and gradually restore spiking activity back to baseline levels (*Turrigiano, 2012*; *Turrigiano, 2008*). Importantly, cytosolic calcium is itself an upstream barometer of activity that regulates the molecular signaling cascades underlying AMPA receptor scaling or GABA receptor removal (*Harris and Rubel, 2006*; *Turrigiano, 2012*). This underscores a key advantage to using calcium imaging in experiments that monitor network activity following perturbations of peripheral input levels: that although GCaMP is a closely related but indirect measure of spiking, it arguably provides more direct insight into a key upstream signal driving Homeostatic Plasticity signaling cascades.

Although we did not measure Homeostatic Plasticity per se, via direct demonstrations of intrinsic or heterosynaptic electrophysiological changes, our measurements of spontaneous and sound-evoked calcium transients clearly demonstrate a failure of homeostatic regulation following acoustic trauma. Spontaneous and sound-evoked calcium levels both remained elevated above pre-exposure baseline levels or levels observed in sham-exposed control mice. Hyperactive and hyper-correlated activity in regions of the cortical topographic map corresponding to peripheral sensorineural damage is widely understood to be the underlying neural basis of phantom sound perception in tinnitus (*Auerbach et al., 2019*; *Herrmann and Butler, 2021b*; *Noreña, 2011*). Here, we show that these changes are also a likely underlying neural substrate for auditory hypersensitivity, a core component of hyperacusis that often accompanies tinnitus (*Cederroth et al., 2020*; *Schecklmann et al., 2014*). These findings identify a neurophysiological target for testing therapeutic strategies in animals and inform the selection of non-invasive biomarkers for the development of improved diagnostics and therapies in human populations (*Polley and Schiller, 2022*).

## Materials and methods

### Key resources table

| Reagent type (species) or resource | Designation | Source or reference | Identifiers | Additional information |
|---|---|---|---|---|
| Genetic reagent (*Mus musculus*) | C57BL/6J-Tg(Thy1-GCaMP6s) GP4.12Dkim/J | Jackson Laboratory | JAX #025776 | Male |
| Genetic reagent (*Mus musculus*) | CBA/CaJ | Jackson Laboratory | JAX #000654 | Female |

*Continued on next page*

*Continued*

| Reagent type (species) or resource | Designation | Source or reference | Identifiers | Additional information |
|---|---|---|---|---|
| Genetic reagent (*Mus musculus*) | C57BL/6J | Jackson Laboratory | JAX #000664 | Male/female |
| Recombinant DNA reagent | AAVrg-pgk-Cre | Addgene | RRID:Addgene_24593 | |
| Recombinant DNA reagent | AAV5-Ef1a-DIO hChR2(E123T/T159C)-EYFP | Addgene | RRID:Addgene_35509 | |
| Antibody | ms(1gG1) α CtBP2 (mouse monoclonal) | BD Transduction Labs | BDB612044 | 1:200 |
| Antibody | rb α MyosinVIIa (rabbit polyclonal) | Proteus Biosciences | 25–6790 | 1:200 |
| Antibody | ms(1gG2a) α GluA2 (mouse monoclonal) | Millipore | MAB397 | 1:2000 |
| Antibody | rb α Epsin (rabbit polyclonal) | Sigma | HPA028674 | 1:100 |
| Antibody | gt α ms (IgG2a) AF 488 (goat polyclonal) | Thermo Fisher | A-21131 | 1:1000 |
| Antibody | gt α ms (IgG1) AF 568 (goat polyclonal) | Thermo Fisher | A-21124 | 1:1000 |
| Antibody | dk α rb AF 647 (donkey polyclonal) | Thermo Fisher | A-31573 | 1:200 |
| Antibody | gt α rb PacBlue (goat polyclonal) | Thermo Fisher | P10994 | 1:200 |
| Chemical compound, drug | Lidocaine hydrochloride | Hospira Inc | Cat #71-157-DK | |
| Chemical compound, drug | Buprenorphine hydrochloride | Buprenex | Cat #NDC 12496-0757-5 | |
| Chemical compound, drug | Isoflurane | Piramal | Cat #NDC 66794-013-10 | |
| Chemical compound, drug | Silicon adhesive | WPI | Cat #KWIK-SIL | |
| Chemical compound, drug | C&B Metabond Quick Adhesive Cement System | Parkwell | Cat #S380 | |
| Software, algorithm | Labview | National Instruments | https://www.ni.com/en-us/shop/labview.html | Version 2015 |
| Software, algorithm | ThorImage 3.0 | Thorlabs | https://www.thorlabs.com/newgrouppage9.cfm?objectgroup | |
| Software, algorithm | Suite2P | Github | https://github.com/cortex-lab/Suite2P; **The Cortical Processing Laboratory at UCL, 2019** | **Pachitariu et al., 2016** |
| Software, algorithm | CellReg | Github | https://github.com/zivlab/CellReg; **zivlab, 2022** | **Sheintuch et al., 2017** |
| Software, algorithm | MATLAB | Mathworks | https://www.mathworks.com/ products/matlab.html | Version 2017b |
| Other | Solenoid driver | Eaton-Peabody Labs | https://github.com/EPL-Engineering/epl_valve; **EPL-Engineering, 2019b** | See Methods, 'Go/NoGo tone detection task' |
| Other | Lickometer | Eaton-Peabody Labs | https://github.com/EPL-Engineering/epl_lickometer; **EPL-Engineering, 2019a** | See Methods, 'Go/NoGo tone detection task' |
| Other | PXI Controller | National Instruments | PXIe-8840 | See Methods, 'Go/NoGo tone detection task' |

*Continued on next page*

*Continued*

| Reagent type (species) or resource | Designation | Source or reference | Identifiers | Additional information |
|---|---|---|---|---|
| Other | Free-field speaker | Parts Express | 275-010 | See Methods, 'Go/NoGo tone detection task' |
| Other | Ti-Sapphire Laser | Spectra Physics | Mai Tai HP DeepSee | See Methods, 'Widefield and two-photon calcium imaging' |
| Other | ×16/0.8 NA Objective | Nikon | CFI75 LWD 16X W | See Methods, 'Widefield and two-photon calcium imaging' |
| Other | Two-photon microscope | Thorlabs | Bergamo II | See Methods, 'Widefield and two-photon calcium imaging' |
| Other | Titanium headplate | iMaterialise | Custom | See Methods, 'Survival surgeries for awake, head-fixed experiments' |
| Other | Diode laser (488 nm) | Omnicron | LuxX 488-100 | See Methods 'Go/NoGo optogenetic detection task' |

## Experimental model and subject details

All procedures were approved by the Massachusetts Eye and Ear Animal Care and Use Committee and followed the guidelines established by the National Institutes of Health for the care and use of laboratory animals. Imaging and tone Go/NoGo behavior were performed on Thy1-GCaMP6s × CBA mice. Combined acoustic and optogenetic Go/NoGo behavioral studies were performed in C57BL/6J mice. Mice of both sexes were used for this study. Noise exposure occurred at 9 weeks postnatal and was timed to occur in the morning, when the temporary component of the threshold shift is less extreme and variable (*Meltser et al., 2014*). Mice were maintained on a 12 hr light/12 hr dark cycle. Mice were provided with ad libitum access to food and water unless they were on-study for behavioral testing, in which case they had restricted access to water in the home cage.

Data were collected from 44 mice. A total of 22 mice contributed data to behavioral tasks: 13 ($N$ = 7/6, trauma/sham) to the tone Go/NoGo behavior and 9 ($N$ = 6/3, trauma/sham) to the combined acoustic and optogenetic Go/NoGo behavior. A total of 10 mice contributed to the chronic imaging experiments: two mice were used for widefield imaging ($N$ = 1/1, trauma/sham) and 8 ($N$ = 4/4, trauma/sham) for the two-photon imaging. Twelve mice were only used for regular ABR testing after acoustic trauma to determine the progression of threshold shift. Cochlear histology was performed on 11 of the mice used for Go/NoGo behavioral testing ($N$ = 7/4, trauma/sham). The timing of all procedures performed on each mouse is provided in *Supplementary file 1*.

## Method details

### Survival surgeries for awake, head-fixed experiments

Mice were anesthetized with isoflourane in oxygen (5% induction, 1.5–2% maintenance). A homeothermic blanket system was used to maintain body temperature at 36.6 (FHC). Lidocaine hydrochloride was administered subcutaneously to numb the scalp. The dorsal surface of the scalp was retracted and the periosteum was removed. The skull surface was prepped with etchant and 70% ethanol before affixing a titanium headplate to the dorsal surface with dental cement. At the conclusion of the head-plate attachment and any additional procedures listed below, Buprenex (0.05 mg/kg) and meloxicam (0.1 mg/kg) were administered, and the animal was transferred to a warmed recovery chamber.

### High-frequency noise exposure

To induce acoustic trauma, octave-band noise at 16–32 kHz was presented at 103 dB SPL for 2 hr. Exposure stimulus was delivered via a tweeter fixated inside a custom-made exposure chamber (51 × 51 × 51 cm). The interior walls of the acoustic enclosure joined at irregular, non-right angles to minimize standing waves. Additionally, to further diffuse the high-frequency sound field, irregular surface depths were achieved on three of the interior walls by attaching stackable ABS plastic blocks (LEGO).

Prior to exposure, mice were placed, unrestrained, in an independent wire-mesh chamber (15 × 15 × 10 cm). This chamber was placed at the center of a continuously rotating plate, ensuring mice were exposed to a relatively uniform sound field. Sham-exposed mice underwent the same procedure except that the exposure noise was presented at an innocuous level (40 dB SPL). All sham and noise exposures were performed at the same time of day.

## Go/NoGo tone detection task

Three days after headplate surgery, animals were weighed and placed on a water restriction schedule (1 ml/day). During behavioral training, animals were weighed daily to ensure they remained between 80% and 85% of their initial weight and regularly examined for signs of excess dehydration. Mice were given supplemental water if they received less than 1 ml during a training session or appeared excessively dehydrated. During testing, mice were head-fixed in a dimly lit, single-walled sound attenuating booth (ETS-Lindgren), with their bodies resting in an electrically conductive cradle. Tongue contact on the lickspout closed an electrical circuit that was digitized (at 40 Hz) and encoded to calculate lick timing. Digital and analog signals controlling sound delivery and water reward were controlled by a PXI system with custom software programmed in LabVIEW. Free-field stimuli were delivered via an inverted dome tweeter positioned 10 cm from the left ear and calibrated with a wide-band ultrasonic acoustic sensor (Knowles Acoustics).

Most mice required 2 weeks of behavioral shaping before they could perform the complete tone detection task with psychophysical staircasing. After mice were habituated to head-fixation, they were conditioned to lick the spout within 2 s following the onset of an 8 or 32 kHz 70 dB SPL tone (0.25 s duration, with 5 ms raised cosine onset–offset ramps) to receive a small quantity of water (4 μl). Trials had a variable intertrial interval (4–10 s) randomly selected from a truncated exponential distribution. Once reaction times were consistently <1 s, mice were trained to detect 8 and 32 kHz tones in a 2-down, 1-up adaptive staircasing paradigm, where two correct detections were required to decrease the range of sound intensities by 5 dB SPL and one miss was required to increase the range of sound intensities by 5 dB SPL. At each iteration of the adaptive staircasing procedure, three trials were presented: a catch (silent) trial and tones at ±5 dB SPL relative to the last intensity tested (*Figure 1I*). A single frequency was presented until 1 reversal was reached, and then the other tone was presented; a run was completed once six reversals had been reached for both frequencies. The first frequency presented on each daily session was randomized.

Hits were defined as post-target licks that occurred >0.1 and <1.5 s following the onset of the target tone. False alarms (Go responses on a catch trial) triggered a 5-s time out. Entire runs were excluded from analysis if the false alarm rate was greater than 30%. This exclusion criterion resulted in the elimination of <5% of test runs across all conditions (before, after, noise- or sham-exposure), underscoring that mice were under stimulus control even if their hearing thresholds were elevated. Psychometric functions were fit using binary logistic regression. Threshold was defined as the average intensity at reversals across an entire session.

## Go/NoGo optogenetic detection task

Headplate attachment, anesthesia and analgesia followed the procedure described above. Three burr holes were made in the skull over auditory cortex (1.75–2.25 mm rostral to the lambdoid suture). We first expressed Cre-recombinase in neurons that project to the ACtx by injecting 150 nl of AAVrg-pgk-Cre 0.5 mm below the pial surface at three locations within the ACtx with a precision injection system (Nanoject III) coupled to a stereotaxic positioner (Kopf). A fourth injection was then performed to selectively express channelrhodopsin in auditory thalamocortical projection neurons by injecting 100 nl of AAV5-Ef1a-DIO hChR2(E123T/T159C)-EYFP in the MGBv (−2.95 mm caudal to bregma, 2.16 mm lateral to midline, 3.05 mm below the pial surface). An optic fiber (flat tip, 0.2 mm diameter, Thorlabs) was inserted at the MGB injection coordinates to a depth of 2.9 mm below the pial surface. The fiber assembly was cemented in place and painted with black nail polish to prevent light leakage. Animals recovered for at least 3 days before water restriction and behavioral testing began.

After mice were habituated to head-fixation, they were conditioned to lick the spout within 2 s following the onset of 70 dB SPL high-frequency bandpass noise (centered at 32 kHz, width 1 octave). Once consistent, mice were trained to detect optogenetic activation of thalamocortical neurons. The laser was pulsed at 10 Hz, 10 ms pulse width for 500 ms, and the bandpass noise was pulsed at 10 Hz,

20 ms pulse width (5 ms raised cosine onset–offset ramps) for 500 ms. For testing, randomized inter-leaved blocks of either noise or laser stimulation were presented at a fixed range of levels. The range of sound levels and laser powers was individually tailored prior to noise/sham exposure to ensure equivalent sampling of sound and laser perceptual growth functions, and then these fixed values were used for all post-exposure testing sessions. Tailoring was accomplished by first identifying the lowest laser power and 32 kHz sound level that produced at least 95% hit rates (operationally defined as 'max'). These sound levels and laser powers were then presented alongside four attenuated levels relative to the maximum as well as no-stimulus catch trials in each mouse on every session. Psychometric functions were fit using binary logistic regression, and threshold was defined as the point where detection crossed 71% correct, which is the closest approximation to the threshold point identified with the 2-up, 1-down staircasing procedure described above. Runs were rejected for further analysis if the false alarm rate of the mouse was above 30%, and again this resulted in the exclusion of <5% of sessions.

## Widefield and two-photon calcium imaging

Three round glass coverslips (two 3 mm and one 4 mm diameter, #1 thickness, Warner Instruments) were etched with piranha solution and bonded into a vertical stack using transparent, UV-cured adhesive (Norland Products, Warner Instruments). Headplate attachment, anesthesia and analgesia follow the procedure listed above. A circular craniotomy (3 mm diameter) was made over the right ACtx using a scalpel and the coverslip stack was cemented into the craniotomy. Animals recovered for at least 5 days before beginning imaging recordings. All imaging was performed in awake, passively listening animals.

A series of pilot widefield imaging experiments were performed to visualize changes in all fields of the ACtx over a longer, 30- to 60-day post-exposure time period (N = 8 noise-exposed and 7 sham-exposed mice). The data collection procedure for these pilot experiments followed the methods described in detail in our previous publication (*Romero et al., 2019*). Briefly, widefield epifluorescence images were acquired with a tandem-lens microscope (THT-microscope, SciMedia) configured with low-magnification, high-numerical aperture lenses (PLAN APO, Leica, ×2 and ×1 for the objective and condensing lenses, respectively). Blue illumination was provided by a light-emitting diode (465 nm, LEX2-LZ4, SciMedia). Green fluorescence passed through a filter cube and was captured at 20 Hz with a sCMOS camera (Zyla 4.2, Andor Technology).

Cellular imaging was performed with a two-photon imaging system in a light-tight sound-attenuating enclosure mounted on a floating table (Bergamo II Galvo-Resonant 8 kHz scanning microscope, Thorlabs). An initial lower resolution epifluorescence widefield imaging session was performed with a CCD camera to visualize the tonotopic gradients of the ACtx and identify the position of A1 (as shown in *Figure 3C*). Two-photon excitation was provided by a Mai-Tai eHP DS Ti:Sapphire-pulsed laser tuned to 940 nm (Spectra-Physics). Imaging was performed with a ×16/0.8 NA water-immersion objective (Nikon) from a 512 × 512 pixel field of view at 30 Hz with a Bergamo II Galvo-Resonant 8 kHz scanning microscope (Thorlabs). Scanning software (Thorlabs) was synchronized to the stimulus generation hardware (National Instruments) with digital pulse trains. The microscope was rotated by 50–60 degrees off the vertical axis to obtain images from the lateral aspect of the mouse cortex while the animal was maintained in an upright head position. Animals were monitored throughout the experiment to confirm all imaging was performed in the awake condition using modified cameras (PlayStation Eye, Sony) coupled to infrared light sources. Imaging was performed in layers L2/3, 175–225 mm below the pial surface. Fluorescence images were captured at ×1 digital zoom, providing an imaging field of 0.84 × 0.84 mm.

Raw calcium movies were processed using Suite2P, a publicly available two-photon calcium imaging analysis pipeline (*Pachitariu et al., 2016*). Briefly, movies are registered to account for brain motion. Regions of interest are established by clustering neighboring pixels with similar time courses. Manual curation is then performed to eliminate low quality or non-somatic regions of interest. Spike deconvolution was also performed in Suite2P, using the default method based on the OASIS algorithm (*Friedrich et al., 2017*). For chronic tracking of individual cells across imaging sessions, cross-day image registration was performed using a method outlined by *Sheintuch et al., 2017*. Briefly, fields-of-view are aligned to a reference imaging session using a non-rigid transformation, and a probabilistic modeling approach is used to estimate whether neighboring cells from separate sessions are

the same or different cells. To estimate the false positive rate with this approach, we also performed a control in which cross-day registration was performed with daily imaging fields randomly selected from different mice (*Figure 6—figure supplement 1*). For all analysis of tracked neurons, only cells with a confidence score of at least 0.8 (max of 1) and that were tracked for at least 8 of the 15 imaging sessions were used for the analysis. In analyses identifying cells that were either 'lost' or 'appeared' after noise/sham exposure (*Figure 6—figure supplement 1*), 'appeared' cells were defined as not being found in any baseline imaging sessions and also found in at least eight imaging sessions after noise exposure. Cells 'lost' at day X relative to noise exposure were consecutively tracked in all baseline sessions and days 0 through X, and not found in any session after day X. The same confidence threshold of 0.8 was also applied to analyses of 'lost' and 'appeared' cells.

During widefield imaging sessions, 20–70 dB SPL tones (in 10 dB steps) were presented from 4 to 64 kHz in 0.5 octave steps. On the first and last two-photon imaging sessions and on the day of noise exposure, 20–80 dB SPL tones (15 dB steps) were presented from 4 to 45.3 kHz (0.5 octave steps). For all other two-photon imaging sessions, 20–80 dB SPL tones (in 10 dB steps) were presented at 5.7, 8, 11.3, and 32 kHz. Each day, all stimuli were repeated 20 times. One block consisted of all frequency–intensity combinations, and stimuli were randomized within blocks. Tones were 50 ms with 5 ms raised cosine onset–offset ramps with 3 s inter-trial intervals.

## Cochlear function tests

Animals were anesthetized with ketamine (120 mg/kg) and xylazine (12 mg/kg), were placed on a homeothermic heating blanket during testing, with half the initial ketamine dose given as a booster when required. Acoustic stimuli were presented via in-ear acoustic assemblies consisting of two miniature dynamic earphones (CUI CDMG15008-03A) and an electret condenser microphone (Knowles FG-23339-PO7) coupled to a probe tube. To highlight the peripheral neural contribution to the ABR, subdermal electrodes were positioned in a horizontal (pinna-to-pinna) montage (*Galbraith et al., 2006*; *Melcher et al., 1996*). Stimuli were calibrated in the ear canal in each mouse before recording. ABR stimuli were 5 ms tone pips at 8, 12, 16, or 32 kHz with a 0.5 ms rise-fall time delivered at 30 Hz. Intensity was incremented in 5 dB steps, from 20 to 100 dB SPL. ABR threshold was defined as the lowest stimulus level at which a repeatable waveform could be identified. DPOAEs were measured in the ear canal using primary tones with a frequency ratio of 1.2, with the level of the f2 primary set to be 10 dB less than f1 level, incremented together in 5 dB steps. The 2f1–f2 DPOAE amplitude and surrounding noise floor were extracted. DPOAE threshold was defined as the lowest of at least two continuous f2 levels, for which the DPOAE amplitude was at least two standard deviations greater than the noise floor. DPOAE and ABR testing was performed 1 week before noise- or sham-exposure, and again immediately following the conclusion of behavioral testing or imaging.

## Cochlear histology

To visualize cochlear afferent synapses and IHCs and OHCs, cochleae were dissected and perfused through the round window and oval window with 4% paraformaldehyde in phosphate-buffered saline (PBS), then post-fixed in the same solution for 1 hr. Cochleae were then decalcified in 0.12 M ethylenediaminetetraacetic acid (EDTA) for 2 days and dissected into half-turns for whole-mount processing. Immunostaining began with a blocking buffer (PBS with 5% normal goat or donkey serum and 0.2–1% Triton X-100) for 1 hr at room temperature. Whole mounts were then immunostained by incubating with a combination of the following primary antibodies: (1) rabbit anti-CtBP2 at 1:100, (2) rabbit anti-myosin VIIa at 1:200, and (3) mouse anti-GluR2 at 1:2000 and secondary antibodies coupled to the red, blue, and green channels. Immunostained cochlear pieces were measured, and a cochlear frequency map was computed (*Müller et al., 2005*) to associate structures to relevant frequency regions using a plug-in to ImageJ (*Parthasarathy and Kujawa, 2018*).

Images were collected at 2400 × 900 raster using a using a high-resolution, oil immersion objective (×63, numerical aperture 1.3), and ×1.25 zoom and assessed for signs of damage. Confocal z-stacks at identical frequencies were collected using a Leica TCS SP5 microscope to visualize hair cells and synaptic structures. Two adjacent stacks were obtained (78 µm cochlear length per stack) at each target frequency spanning the cuticular plate to the synaptic pole of ~10 hair cells (in 0.25 µm z-steps). Images were collected in a 1024 × 512 raster using a high-resolution, oil immersion objective (×63, numerical aperture 1.3), and digital zoom (×3.17). Images were loaded into an image-processing

software platform (Amira; VISAGE Imaging), where IHCs were quantified based on their Myosin VIIa-stained cell bodies and CtBP2-stained nuclei. Presynaptic ribbons and postsynaptic glutamate receptor patches were counted using 3D representations of each confocal z-stack. Juxtaposed ribbons and receptor puncta constitute a synapse, and these synaptic associations were determined by calculating and displaying the *x–y* projection of the voxel space within 1 μm of each ribbon's center (*Liberman et al., 2011*). OHCs were counted based on the myosin VIIa staining of their cell bodies. The mean number of cells per row of OHCs was used as a measure of OHC counts.

For visualizing OHC stereocilia damage, following similar whole-mount dissection and blocking procedures, the other ear was immunostained with a combination of the following primary antibodies (1) rabbit anti-CtBP2 at 1:100, (2) mouse anti-GluR2 at 1:2000, and (3) rabbit anti-Espin at 1:100, followed by secondary antibodies in the red, green, and gray channels. Confocal z-stacks of the stereocilia were collected at 5.6, 11.3, 22, 32, 45, and 64 kHz cochlear frequencies with a Leica TCS SP8 microscope.

### Brain histology

For mice performing the Go/NoGo optogenetic detection task, mice were deeply anesthetized and prepared for transcardial perfusion with a 4% formalin solution in 0.1 M phosphate buffer 21 days after noise exposure. The brains were extracted and post-fixed at room temperature for an additional 12 hr before transfer to 30% sucrose solution. Coronal sections (50 μm) were mounted onto glass slides using Vectashield with DAPI, and then coverslipped. Regions of interest were then imaged at ×10 using a Leica DM5500B fluorescent microscope.

## Quantification and statistical analysis

### Clinical database analysis

First-visit patient records from the Massachusetts Eye and Ear audiology database over a 24-year period from 1993 to 2016 were analyzed. Our analysis selected for adult patients aged 18 and 80, whose primary language was English, and who underwent pure tone audiogram tests in the left and right ears with octave spaced frequencies between 250 Hz and 8 kHz using headphones or inserts. To eliminate patients with conductive components in their hearing loss, the MEE dataset was further curated to remove all audiograms where the air-bone gap was ≥20 dB at any one frequency or ≥15 dB at two consecutive frequencies. Audiograms with thresholds ≥85 dB HL at frequencies ≤2000 Hz were also removed to maintain a conservative inclusion criterion, as the difference in limits of the air and bone conducting transducers limit our ability to determine the presence of conductive components in that threshold range. After this exclusionary step, we were left with 132,504 audiograms in the dataset for analysis. Of these audiograms, HFHL was defined as audiograms with thresholds lower than 20 dB HL for frequencies <1 kHz, between 10 and 80 dB HL for 2 kHz, between 20 and 120 dB HL for 4 kHz, and between 40 and 120 dB HL for 8 kHz, following the same criteria used to identify a steeply sloping high-frequency hearing loss in prior clinical database studies (*Dubno et al., 2013*; *Parthasarathy et al., 2020b*; *Vaden et al., 2017*). HFHL audiometric phenotypes consisted of 23% of the audiograms assessed based on these criteria. These patients were 65% male and had a median age of 65 years. The study was approved by the human subjects Institutional Review Board at Mass General Brigham and Massachusetts Eye and Ear. Data analysis was performed on deidentified data, in accordance with the relevant guidelines and regulations.

### Behavioral analysis

To estimate perceptual gain in the Go/NoGo tone detection task, hit rates were taken at intensities ranging from the lowest intensity with sufficient trials (>5 trials) to the first intensity at which the hit rate was above 90% to account for saturation of the detection function. The perceptual gain was calculated as the average first derivative of the *d′* function evaluated over the specified intensity range. The gain calculation was performed for each animal based on aggregated daily test sessions for a given post-exposure epoch (e.g., baseline, 0–2 days post-exposure).

### Two-photon image analysis

All analysis was performed on the deconvolved calcium activity traces. For analysis of tone-evoked responses, averaged deconvolved calcium traces were expressed as *Z*-score units relative to activity

levels measured during the pre-stimulus period (833 ms). PyrNs were operationally defined as being responsive to a particular tone frequency/level combination with a $Z > 2$. For the three imaging sessions that calculated the full frequency-response area, the minimum response threshold was defined for each PyrN as the lowest level at which there were responses to two adjacent frequencies (frequencies 0.5 octaves apart). Best frequency (BF) was defined as the frequency for which the overall response was maximal over the intensity range of threshold + 30 dB. Analysis of BF changes was limited to PyrNs with pure tone receptive fields (neural $d' > 1$, as defined in *Romero et al., 2019*).

To delineate the intact and deafferented regions of the imaging field, an SVM was calculated for each mouse using BF's determined from the first day of imaging prior to noise exposure. The SVM deafferentation boundary categorized physical space (intact: BF <16 kHz, deafferented: BF ≥16 kHz) and its physical location was then imposed on all successive imaging sessions after alignment of fields-of-view from all imaging sessions. To categorize the position of a neuron, the adjusted centroid locations were used based on the best registration from the previously mentioned method. The distance of a neuron to the SVM deafferentation boundary was calculated as the shortest Euclidean distance.

Gain was defined as the relationship between sound level (input) and activity rate (output). The gain was calculated as the average rate of change over a range of sound levels. The particular set of sound levels selected for gain analysis was determined according to whether the best level occurred at low, mid, or high sound levels as illustrated in *Figure 5—figure supplement 1*. For a neuron to be considered for an analysis of gain, it was required to have a significant response ($Z > 2$) to at least three consecutive intensities.

Spontaneous activity was calculated from the 833 ms periods preceding tone onset. To quantify the correlated activity between cells, we cross-correlated the $Z$-scored activity in the pre-stimulus periods. To control for the effects of overall changes in activity rates over sessions or between cells, shuffled cross-correlograms were generated for each pair by shuffling trial labels. Only cross-correlograms for which at least three consecutive lags had values significantly greater than the shuffled cross-correlogram (bootstrapped p < 0.05, Bonferroni corrected for multiple comparisons) were used for analysis. The degree of correlated activity between each pair was defined as the size of the positive area under the peak of the shuffle-subtracted cross-correlogram (xcorr area).

To determine how ensemble activity decoded tone presence, we used an SVM classifier with a linear kernel (following the approach of *Resnik and Polley, 2021*). The SVM was run on the principal components of a data matrix consisting of the $Z$-scored responses to single tone presentations or silent periods. PCA was used to reduce the influence of any inequities in sample sizes across mice or conditions. We ran the SVM on the minimum number of principal components required to explain 90% of the variance. Leave-one-out cross-validation was then used to train the classifier and compute the decoder accuracy. We repeated this process independently for each frequency intensity at 8 and 32 kHz for each imaging session. The models were fit using the 'fitcsvm' function in MATLAB.

To model how pre-exposure properties can predict the change in a neuron's responsiveness, the outcome variable was the post/pre-exposure ratio of the areas under the intensity-response growth function, where post was drawn from days 3 to 5 after exposure. All predictor variables were computed as average values from the pre-exposure period. The best linear model was fit using stepwise multiple linear regression and using the Akaike information criterion. For the purposes of comparison, the predictor variables from the trauma model were applied to the sham model. The stepwise regression was fit using 'stepwiselm' and subsequent model fits used 'fitlm' in MATLAB.

## Statistical analysis
All statistical analyses were performed in MATLAB 2017b (Mathworks). Data are reported as mean ± standard error of the mean unless otherwise indicated. Post hoc pairwise comparisons were adjusted for multiple comparisons using the Bonferroni correction.

## Acknowledgements
These studies were supported by a grant from the Nancy Lurie Marks Family Foundation (DP and AT), NIH grant DC009836 (DP), DC015857 (DP and SK), DC018353 (AT), and NIH fellowship DC018974-02 (MM) and DC014871 (AH). We thank MC Liberman and A Indzhykulian for their assistance with hair cell stereocilia imaging.

## Additional information

### Funding

| Funder | Grant reference number | Author |
| --- | --- | --- |
| National Institute on Deafness and Other Communication Disorders | DC018974-02 | Matthew McGill |
| National Institute on Deafness and Other Communication Disorders | DC014871 | Ariel E Hight |
| Nancy Lurie Marks Family Foundation | | Anne Takesian Daniel B Polley |
| National Institute on Deafness and Other Communication Disorders | DC009836 | Daniel B Polley |
| National Institute on Deafness and Other Communication Disorders | DC015857 | Sharon G Kujawa Daniel B Polley |
| National Institute on Deafness and Other Communication Disorders | DC018353 | Anne Takesian |

The funders had no role in study design, data collection, and interpretation, or the decision to submit the work for publication.

### Author contributions

Matthew McGill, Conceptualization, Data curation, Formal analysis, Funding acquisition, Methodology, Writing - original draft, Writing – review and editing; Ariel E Hight, Conceptualization, Data curation, Formal analysis, Writing – review and editing; Yurika L Watanabe, Dongqin Cai, Data curation, Writing – review and editing; Aravindakshan Parthasarathy, Data curation, Formal analysis, Writing – review and editing; Kameron Clayton, Methodology, Writing – review and editing; Kenneth E Hancock, Software, Methodology; Anne Takesian, Sharon G Kujawa, Supervision, Writing – review and editing; Daniel B Polley, Conceptualization, Supervision, Writing - original draft, Writing – review and editing

### Author ORCIDs

Matthew McGill http://orcid.org/0000-0003-2322-9580
Daniel B Polley http://orcid.org/0000-0002-5120-2409

### Ethics

The study was approved by the human subjects Institutional Review Board at Mass General Brigham and Massachusetts Eye and Ear. Data analysis was performed on deidentified data, in accordance with the relevant guidelines and regulations.

All procedures were approved by the Massachusetts Eye and Ear Animal Care and Use Committee and followed the guidelines established by the National Institutes of Health for the care and use of laboratory animals.

### Decision letter and Author response

Decision letter https://doi.org/10.7554/eLife.80015.sa1
Author response https://doi.org/10.7554/eLife.80015.sa2

## Additional files

### Supplementary files

• Supplementary file 1. Timing of all procedures performed in each mouse.

• MDAR checklist

## Data availability

All figure code and data will be available on the Harvard Dataverse at the following: https://doi.org/10.7910/DVN/JLIKOZ.

The following dataset was generated:

| Author(s) | Year | Dataset title | Dataset URL | Database and Identifier |
| --- | --- | --- | --- | --- |
| McGill M | 2022 | Neural signatures of auditory hypersensitivity following acoustic trauma | https://doi.org/10.7910/DVN/JLIKOZ | Harvard Dataverse, 10.7910/DVN/JLIKOZ |

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
