## [Editor Report]

This is an important and methodologically compelling paper that provides novel views on the functional plasticity of cortical networks following noise trauma. The combination of cortical recordings, optogenetics, and behavior provides valuable mechanistic insights into the processes that may underlie auditory hypersensitivity after cochlear damage. The extensive and well-illustrated manuscript provides an excellent example of a study in which modern neurophysiology techniques advance comprehension of pathologies related to maladaptive changes in the brain.

---

## [Decision Letter]

**Decision letter after peer review:**

Thank you for submitting your article "Neural signatures of auditory hypersensitivity following acoustic trauma" for consideration by *eLife*. Your article has been reviewed by 3 peer reviewers, including Brice Bathellier as Reviewing Editor and Reviewer #1, and the evaluation has been overseen by Barbara Shinn-Cunningham as the Senior Editor. The following individuals involved in the review of your submission have agreed to reveal their identity: Arnaud Norena (Reviewer #2); Victoria M Bajo Lorenzana (Reviewer #3).

Essential revisions:

1) Please either address the concerns of reviewers 2 and 3 about the behavioral readout of hyperacusis or tone down the conclusions to preclude over-interpreting indirect measurements of perception. In particular, not all animals experiencing noise trauma will experience hyperacusis. One possible way to improve the description of the hyperacusis condition would be to show a correlation between behavioral results and physiological measurement at the single animal level.

2) The referees have requested a large number of clarifications. Please carefully take this into account.

*Reviewer #1 (Recommendations for the authors):*

I have no particular comment on this manuscript which, to me, is carefully written, and includes carefully executed experiments and interesting results.

*Reviewer #2 (Recommendations for the authors):*

Specific comments

"As an exception, homeostatic regulation of neural activity often fails in the adult auditory system after hearing loss" – what do you mean?

"ii) a selective inability to perceptually suppress distracting sensory features,… "

I would remove this aspect as it is not really addressed in the paper and it is (in my opinion) different from tinnitus and hyperacusis: it is a more high-level phenomenon linked to informational masking.

"volume of electrical activity"

The use of "volume" is awkward. Why not use "surface"? I would use another term (level, sum, integral? etc.).

"These changes are often described as homeostatic plasticity, but in the context of adult plasticity after hearing loss they reflect a failure in homeostatic processes that maintain neural excitability at a stable set point. Whereas homeostatic changes – by definition – restore neural activity to a baseline activity rate following a perturbation (Turrigiano, 2012), neural gain adjustments following adult-onset hearing loss often over-shoot the mark, producing catastrophic downstream consequences at the level of network excitability and sound perception (Eggermont, 2017; Noreña, 2011)."

See my point in the Public Review. I am not sure we know whether the averaged central activity is really enhanced after hearing loss. What is enhanced is the spontaneous activity and the stimulus-evoked activity for supra-threshold signals, but overall the averaged activity should not be increased. The enhanced neural activity post-hearing loss just compensates for the reduction of stimulus-evoked activity due to hearing loss.

"A closer inspection of the mouse psychometric detection functions for the spared low-frequency tone suggested something akin to the clinical phenomenon of loudness recruitment." I am sorry it is not clear to me.

"Following acoustic trauma, the behavioral sensitivity index (d-prime, or d') for 8 kHz tones grew more steeply for sound intensities near thresholds compared both to pre-exposure baseline detection functions or sham exposure controls (Figure 2A-B)."

The authors should be more explicit since I don't know where to look at them. I don't understand how the sensation level (dB SL) can be negative. Does negative dB SL mean below the threshold? What are the results for high frequencies?

"That intensity-detection slopes were increased for the low-frequency tone at near-threshold sound levels argues against a peripheral origin for this change and instead suggests increased gain in the central auditory pathway." Why? Please provide complete argumentation for this.

"Importantly, testing in interleaved trials revealed behaviorally hypersensitivity to direct stimulation of auditory thalamocortical neurons after acoustic trauma (Figure 2E), demonstrating significantly increased d' across optogenetic stimulation intensities compared to sham controls (Figure 2F)". I love that experiment, testing neural sensitivity independently from the cochlea and hearing loss. However, I don't see much effects on the psychometric function in Figure 2E.

"A marked increase in PyrNs tuned to lesion edge frequencies could contribute to the enhanced perceptual sensitivity to 8 kHz tones identified in behavioral experiments (Figure 2)." Where does this hypothesis come from? Usually, loudness hyperacusis is not frequency-specific and is often found in subjects with small, if any, hearing loss.

Could you explain how the gain is calculated? In Figure 5A left example, I see an absolute response of 10 (15-10) over 40 dB, 10/40=0.4? But the value that is indicated is 0.23. On the other hand, it seems to work on the right example: 65/40=1.62.

"Classification of single trial A1 ensemble responses supported the hypothesis that cortical discrimination of sound versus silence would be enhanced for low- and mid-intensity 8 kHz tones but reduced for 32 kHz tones after trauma (Figure 5E). Enhanced neural detection of 8 kHz tones was largely driven by PyrNs in deafferented map regions, whereas the loss of cortical sensitivity to high-frequency tones was observed in both topographic zones (Figure 5F)."

I don't understand the point here, and more precisely the link between this result and the putative loudness hyperacusis. According to human data, loudness hyperacusis should be found at 8kHz and 32 kHz for supra-threshold acoustic simulation.

Also, it seems that neural activity is unchanged after noise trauma in the "intact region" when neural activity is assessed by calcium imaging, while the activity has been found to be largely enhanced in that region when neural activity is assessed with electrophysiology (MUA and LFP). That should be mentioned in the discussion. I am not a specialist in calcium imaging but the authors may explain somewhere (briefly) the relationship between calcium responses and MUA and LFP. Is there any correlation between calcium responses (the max) and MUA response and LF amplitude?

Could the authors explain how they assess spontaneous activity from calcium imaging? What is the shape of the signal that is recorded? Is this signal similar to spontaneous LFP? Spiking activity? This is important when authors assess the synchrony between cells, and then understand what is synchronized. If the activity is globally increased we expect a "mechanic" increase in synchrony that is difficult to correct. With spiking activity, it is different (it is easier to correct, except for onset and offset responses) since the time duration of a spike is very short compared to the inter-spike interval.

Discussion

I suggest being much more cautious in the interpretation of homeostatic plasticity and behavioral data.

"Furthermore, to control for any changes in licking behavior due to acoustic trauma, our slope measurements are taken from functions using a sensitivity index (d'), which normalizes lick probability according to the false alarm rate determined from the delivery of catch (silent) trials. Thus, a steeper relationship between increasing tone intensity and perceptual sensitivity strongly suggests a hypersensitivity to sound following acoustic trauma (Figure 2)."

I am not convinced by this conclusion. At the threshold, enhanced arousal (due to stress after noise trauma) may increase d'. I am not saying stress is necessarily enhanced in exposed animals but it is just to mention that the link between psychometric function for thresholds and loudness is not as straightforward as the authors seem to believe it is.

The discussion should come back a bit more to the literature and emphasize in more detail what is consistent with the literature, what is not, what is new, etc.

I suggest publishing the behavioral data elsewhere, and to focus here on the neural responses before and after the noise trauma, emphasizing clearly what is new, etc.

*Reviewer #3 (Recommendations for the authors):*

Specific concerns:

Introduction

The communalities between tinnitus, hyperacusis and speech in noise impairment are unknown, and the association between those and hearing loss are known but not what happens exactly at neural level. It seems extreme to put together normal aging with ADHD, autism, or schizophrenia, and claim that tinnitus, hyperacusis, and speech in noise impairment can be involved in all those conditions.

Line 76: we don't know yet whether neural gain adjustments following adult-onset hearing loss overshoot the mark. In fact, Koops, Renken, Lanting and van Dijk claimed that cortical tonotopic map changes in humans are larger in hearing loss than in additional tinnitus. J Neurosci. 2020 Apr 15; 40(16): 3178-3185. doi: 10.1523/JNEUROSCI.2083-19.2020

Please stay consistent with the use of trauma, acoustic trauma, sound overexposure, and high-frequency sound exposure throughout the manuscript.

Results

A clarification about the percentage of animals that develop loudness hypersensitivity after sound overexposure is necessary. As previously mentioned hyperacusis and tinnitus are not happening to all animal models of sensorineural hearing loss overexposed to high-frequency sounds. How the authors distinguish between animals with sensorineural hearing loss and animals with associated tinnitus and/or hyperacusis?

Perceptual hypersensitivity following noise-induced high-frequency sensorineural hearing loss.

Please state more precisely the differences between hyperacusis and hypersensitivity to loud sounds.

Explain why only wave 1 was used to explore the auditory thresholds in ABRs when wave 5, for example, would be easy to identify.

It is not clear how far the animals were tested and the survival time of the animals between the hearing loss triggering and the histology showing loss of synaptic contacts in IHCs in the base of the cochlea and death of OHC in the base with stereocilia changes in base and mid coils. In the different figures, there are different days and groups of days depending on the figure. Please clarify.

Figure 1 panels J, K. Change the colours to make them more different. Why there is a threshold elevation at 8 kHz in the first two days?

Behavioural hypersensitivity to cochlear lesion edge frequencies and direct auditory thalamocortical activation after SNHL.

Would it be possible that the differential distribution of damage between the IHCs and OHCs (only base versus base and middle coil) could explain the changes in ABRs, DPOAE and behavioural thresholds?

Please clarify why in the combined optogenetic (ChR2) and auditory stimulation, only a high frequency centred at 32 KHz. Is the optogenetic stimulation affecting all frequency ranges in vMGN or only high frequency?

In Figure 2, panel E it seems that the psychometric curve in the baseline condition for the hearing loss animals is shifted to the right compared with the baseline in the sham condition. It would be easier to compare them if they are placed in the same panel.

Chronic imaging in A1 reveals tonotopic remapping and dynamic spatiotemporal adjustments in neural response gain after acoustic trauma.

The authors mentioned that tonotopic remapping was limited to the deafferented zone where the proportion of layer 2/3 pyramidal neurons preferentially tuned to lesion edge frequencies is more than doubled following acoustic trauma without any systematic change in response. Please specify the lack of systematic change.

The identification of the frequency edge is convincing in the widefield but less convincing at the neural level. How the deafferentation boundary was established?

Please specific the rational for the use of those four specific frequencies (32, 5.7, 8, 11.3)

Cortical hyperresponsivity and increased gain mirrors behavioural hypersensitivity to spared low-frequency inputs.

This section is not particularly convincing and figure 5 is difficult to follow. For example, in panel D it is impossible to distinguish the two different types if sound or silence. Only panel F is clear and easy to follow.

Tracking changes in activity and local synchrony in individual PyrNs over several weeks.

Explain the differences across neurons in the probability to be tracked just for one day or for 15 session. Please explore the possibility that the baseline properties of the neurons impact their tractability in chronic imaging.

The authors mention that following noise overexposure active pyramidal neurons disappeared from the deafferented area, with a permanent increase in response to 8 kHz tome in the deafferented area but not in the low-frequency area. This increase was first for all intensity levels and later only above 50 dB SPL. When this changes happen? After 3 days? When is the beginning of the 2nd stage? Could the authors distinguish between an acute and a chronic phase?

Predicting the degree of excess central gain after trauma in individual PyrNs based on their pre-exposure response features.

Please expand your explanation about potential heterogeneity. I am not certain whether the authors talk about two different groups of pyramidal neurons or a gradient between low to high spontaneous activity at baseline.

Again here the temporal limit is different and they use 3.5 days. Please explain the rationale of that and use always the same time points with justification.

I am curious to know what will be the result of those functions about central gain if other frequencies were used.

Discussion

Please state the differences between the hearing loss in the animal model by acute sound overexposure from the majority of ageing sensorineural hearing loss (presbyacusis) where the changes are more gradual.

Excess central gain is the hypothesized neural substrate for loudness recruitment and generalised hyperacusis, but can the authors think about other potential causes?

Line 353. Typo long spaces in the same line.

Why and what for pyramidal neurons in layers 2-3 with low spontaneous activity and greater responses to low-intensity tones show more stable gain control after trauma? I do not understand what "stochastic variations in inhibitory tone between different microcircuit milieus" means. Is it about the different types and functions of inhibitory interneurons in the cortex? Can neuromodulators such as ACh play a role too?

Can the different types of neurons be distinguished? Again, what happens when the animals only have hearing loss without tinnitus or hyperacusis? Should be possible that the accumulation of one or more than one symptom after hearing loss depends on the proportion of active cells on the baseline?

The authors mention the startle reflex as an indicator of hypersensitivity but usually, animals show a high degree of adaptation to it. Can the sensory component of the hypersensitivity show habituation or sensitization?

Line 439, again many spaces in the sentence.

Line 448. Check typos in references.

Lines 466-468 needs a reference. Again here, the authors are not identifying tinnitus or even hyperacusis or misophonia?

Methods

Give reasons about the time of 9 weeks post-natal when sound overexposure (9 weeks post-natal) and why was done in the morning that corresponds with the period of higher sleep pressure for the mice.

It is difficult to follow the different animals in the experimental design. Please use a table.

Please specify if the whole ventral division of MGB was targeted with the fibre optic or only the high-frequency region. Was the fibre optic present but inactive in MGB for the control experiments?

Line 571-3. The range of sound levels and laser powers were tailored to each mouse prior to noise exposure to ensure equivalent sampling of sound and laser perceptual growth functions. Please expand the explanation by giving further information about differences across cases.

Line 584-585. It is bizarre using a scalpel to do a craniotomy instead of a dental drill.

---

## [Author Response]

Essential revisions:1) Please either address the concerns of reviewers 2 and 3 about the behavioral readout of hyperacusis or tone down the conclusions to preclude over-interpreting indirect measurements of perception. In particular, not all animals experiencing noise trauma will experience hyperacusis. One possible way to improve the description of the hyperacusis condition would be to show a correlation between behavioral results and physiological measurement at the single animal level.

As detailed in the point-by-point rebuttal below, we have addressed the reviewers’ concerns regarding hyperacusis and limitations of the behavior. This was accomplished by scaling back claims that were not supported by the data and explicitly detailing limitations of the current study design, such as measurements of behavior and physiology in separate mice.

2) The referees have requested a large number of clarifications. Please carefully take this into account.

The manuscript has been extensively revised based on these clarification requests as detailed below.

For clarity, the figure changes are as follows:

– In response to Reviewer 3, we have recolored Figure 1J-K to make the lines more discernible.

– In response to Reviewers’ 2 and 3 concerns about the behavior, we have provided individual mouse data. This is shown in Figure 2B and Figure 2F.

– In response to Reviewers’ 2 and 3 comments, we have provided a more representative trauma mouse in Figure 2E.

– In response to Reviewer 3’s question about the probability of cell tracking over time, we have added panel Figure 6 —figure supplement 1E, which shows the progression of cell disappearance over time.

Reviewer #1 (Recommendations for the authors):I have no particular comment on this manuscript which, to me, is carefully written, and includes carefully executed experiments and interesting results.

We thank Reviewer 1 for their careful review and positive evaluation

Reviewer #2 (Recommendations for the authors):Specific comments"As an exception, homeostatic regulation of neural activity often fails in the adult auditory system after hearing loss" – what do you mean?

We agree that this was not the clearest way to introduce the study. The first two sentences of the abstract have been rewritten such that this phrase no longer appears (Pg. 2, Lns 2-5).

"ii) a selective inability to perceptually suppress distracting sensory features,… "I would remove this aspect as it is not really addressed in the paper and it is (in my opinion) different from tinnitus and hyperacusis: it is a more high-level phenomenon linked to informational masking.

We respect that this is a reasonable opinion. But it is also reasonable to postulate that these distinct perceptual manifestations can share some common underlying causes. For example, we have shown that difficulty perceiving auditory targets in background noise is linked to neural hypersynchrony that builds up just prior to target onset (Resnik and Polley, Neuron 2021). Since the beginning of the introduction is broad to draw readers in from diverse backgrounds, we elected to leave this statement.

"volume of electrical activity"The use of "volume" is awkward. Why not use "surface"? I would use another term (level, sum, integral? etc.).

Thanks. We removed this sentence.

"These changes are often described as homeostatic plasticity, but in the context of adult plasticity after hearing loss they reflect a failure in homeostatic processes that maintain neural excitability at a stable set point. Whereas homeostatic changes – by definition – restore neural activity to a baseline activity rate following a perturbation (Turrigiano, 2012), neural gain adjustments following adult-onset hearing loss often over-shoot the mark, producing catastrophic downstream consequences at the level of network excitability and sound perception (Eggermont, 2017; Noreña, 2011)."See my point in the Public Review. I am not sure we know whether the averaged central activity is really enhanced after hearing loss. What is enhanced is the spontaneous activity and the stimulus-evoked activity for supra-threshold signals, but overall the averaged activity should not be increased. The enhanced neural activity post-hearing loss just compensates for the reduction of stimulus-evoked activity due to hearing loss.

Yes, we addressed this point in the response to the public reviews. The underlying mechanisms and common use of the term “homeostatic plasticity” describes a negative feedback process to stabilize the firing rates of individual neurons at a set point. It is true that excess activity in the cortex may (or may not) be offset by equivalent depression of neural activity in the periphery and brainstem but this would be the product – not the driver – of homeostatic processes that work differently in different brain areas. Again, “should” is the operative word here because measurements of the averaged activity across millions of neurons distributed across dozen auditory brain regions in both hemispheres of the brain have never been made. Even among studies that have made simultaneous recordings from cortical and subcortical stations (e.g., Chambers..Polley et al., Neuron 2016) there was no demonstration that the averaged activity was maintained. But, as discussed in the response to the public reviews, the main point of disagreement with the reviewer is that we are describing activity regulation for individual cortical neurons – because the homeostatic mechanisms work at the level of individual cortical neurons – and really make no statement about the overall activity levels of the entire auditory pathway.

"A closer inspection of the mouse psychometric detection functions for the spared low-frequency tone suggested something akin to the clinical phenomenon of loudness recruitment." I am sorry it is not clear to me.

The reviewer is correct; that wording was unclear. We removed that sentence entirely. This section of the Results now reads (on Pg. 5, Lns 137-144),

“For example, we recently reported that human subjects with normal hearing thresholds but asymmetric degeneration of the left and right auditory nerve perceive tones of fixed physical intensity as louder in the ear with poor auditory nerve integrity, particularly for low intensities near sensation level (Jahn et al., 2022). To determine whether mice may be showing evidence of auditory hypersensitivity, we performed a closer inspection of the mouse psychometric detection functions for the spared 8 kHz tone. This analysis confirmed that, following acoustic trauma, the behavioral sensitivity index (d-prime, or d’) grew more steeply for sound intensities near threshold compared both to pre-exposure baseline detection functions and sham exposed controls (Figure 2A-B).”

"Following acoustic trauma, the behavioral sensitivity index (d-prime, or d') for 8 kHz tones grew more steeply for sound intensities near thresholds compared both to pre-exposure baseline detection functions or sham exposure controls (Figure 2A-B)."The authors should be more explicit since I don't know where to look at them. I don't understand how the sensation level (dB SL) can be negative. Does negative dB SL mean below the threshold? What are the results for high frequencies?

The detection sensitivity (d’) is plotted as a function of sensation level for example mice in 2A, which illustrates the steeper growth in the acoustic trauma mouse. The increase in sensitivity with sound intensity is taken as the slope of the function. Panel 2B shows that the slopes became steeper after acoustic trauma but not after a sham exposure. We added a new panel to figure 2B to also show the slope increase for all individual mice.

The detection threshold was defined with a standard procedure as the average intensity at the reversal of the 2-down 1-up adaptive staircasing, which corresponds to an intensity that will be detected on approximately 71% of trials. So, yes, that means we are presenting sound intensities below the level that will be defined as the threshold/0 dB SL. As such, we can plot the d’ for intensities below 0 dB SL. The reviewer can see in Figure 2A that the d’ values from -10 to 0 dB SPL are above zero because mice occasionally detect tones at intensities below the average of the reversals. This is explained on Pg. 18 Lns 581-589.

“Once reaction times were consistently < 1 s, mice were trained to detect 8 kHz and 32 kHz tones in a 2-down, 1-up adaptive staircasing paradigm, where two correct detections were required to decrease the range of sound intensities by 5 dB SPL and one miss was required to increase the range of sound intensities by 5 dB SPL. At each iteration of the adaptive staircasing procedure, three trials were presented: a catch (silent) trial and tones at +/- 5 dB SPL relative to the last intensity tested (Figure 1I). A single frequency was presented until 1 reversal was reached, and then the other tone was presented; a run was completed once 6 reversals had been reached for both frequencies. The first frequency presented on each daily session was randomized.”

"That intensity-detection slopes were increased for the low-frequency tone at near-threshold sound levels argues against a peripheral origin for this change and instead suggests increased gain in the central auditory pathway." Why? Please provide complete argumentation for this.

We have revised the text on Pg. 6 Lns 148-154 to provide a more explicit rationale for this conclusion.

"Persons with sensorineural hearing loss commonly report loudness recruitment, a disproportionately steep growth of loudness with increasing sound level, that has been accounted for by altered basilar membrane mechanics after OHC damage (Oxenham and Bacon, 2003). However, loudness recruitment is most pronounced for frequencies within the range of hearing loss and at intensities well above sensation level (Buus and Florentine, 2002; Moore, 2004). Here, we observed steeper growth of auditory sensitivity for a spared frequency and at intensities close to sensation level, neither of which would be expected of a purely peripheral origin.”

"Importantly, testing in interleaved trials revealed behaviorally hypersensitivity to direct stimulation of auditory thalamocortical neurons after acoustic trauma (Figure 2E), demonstrating significantly increased d' across optogenetic stimulation intensities compared to sham controls (Figure 2F)". I love that experiment, testing neural sensitivity independently from the cochlea and hearing loss. However, I don't see much effects on the psychometric function in Figure 2E.

To address this point, we made three changes to Figure 2: 1) Figure 2E now shows a more representative example of the optogenetic detection functions in an acoustic trauma mouse; 2) we added arrows to Figure 2F to indicate the example sham and trauma from Figure 2E relative to the other mice in the sample; 3) We added a new subpanel to Figure 2F showing the d’ change across the post-exposure period for each individual mouse. In this revised Figure 2E, we can see that detection of thalamocortical stimulation remained unchanged for the sham-exposed mouse and was increased (indicated by an increase in the d’) for the noise-exposed mouse following acoustic trauma. In Figure 2F, we look at the mean change in the detection function evaluated across laser intensities, and we see that after noise exposure but not sham exposure there is an increase in d’, the sensitivity to thalamocortical stimulation.

"A marked increase in PyrNs tuned to lesion edge frequencies could contribute to the enhanced perceptual sensitivity to 8 kHz tones identified in behavioral experiments (Figure 2)." Where does this hypothesis come from? Usually, loudness hyperacusis is not frequency-specific and is often found in subjects with small, if any, hearing loss.

Part of the confusion is related to the terms “hyperacusis” and “loudness”. As addressed in the response to the public reviews, we have removed statements that our experiments model hyperacusis, per se. Further, we removed all claims that our behavioral evidence provides direct insight into the perception of loudness. Rather than “hyperacusis” and “loudness”, the revised manuscript interprets the findings in light of auditory hypersensitivity, which seems appropriate. To the reviewer’s question, we are making the point that if the 8 kHz tone is recruiting activity from a larger number of excitatory neurons, and the response magnitude from individual neurons have also increased, that these changes would plausibly related to behavioral hypersensitivity to the stimulus. In other words, the neural SNR for the stimulus would be higher, which could amount to greater perceptual salience. This seems like a reasonable and non-controversial possibility to raise.

Could you explain how the gain is calculated? In Figure 5A left example, I see an absolute response of 10 (15-10) over 40 dB, 10/40=0.4? But the value that is indicated is 0.23. On the other hand, it seems to work on the right example: 65/40=1.62.

Sure, Pg. 22 Lns 791-795 of the Methods state,

“Gain was defined as the relationship between sound level (input) and activity rate (output). The gain was calculated as the average rate of change over a range of sound levels. The particular set of sound levels selected for gain analysis was determined according to whether the best level occurred at low, mid, or high sound levels as illustrated in Figure 5 —figure supplement 1. “

For the example functions in Figure 5A, gain was evaluated from 40 to 80 dB SPL. For the left function, this is a response change of ~9.2 (a.u.) over 40 dB SPL, which would be a gain of 9.2/40 = 0.23.

"Classification of single trial A1 ensemble responses supported the hypothesis that cortical discrimination of sound versus silence would be enhanced for low- and mid-intensity 8 kHz tones but reduced for 32 kHz tones after trauma (Figure 5E). Enhanced neural detection of 8 kHz tones was largely driven by PyrNs in deafferented map regions, whereas the loss of cortical sensitivity to high-frequency tones was observed in both topographic zones (Figure 5F)."I don't understand the point here, and more precisely the link between this result and the putative loudness hyperacusis. According to human data, loudness hyperacusis should be found at 8kHz and 32 kHz for supra-threshold acoustic simulation.

Part of the confusion stems from the fact that our experiments don’t attempt to model loudness hyperacusis, which, as the reviewer points out, is often observed in human subjects without hearing loss. This was not a major point we tried to emphasize in the original manuscript but the Reviewer is right to point out that there were a few instances where we drew parallels between our findings and hyperacusis. Those comparisons have been removed and, as discussed above, we only draw parallels between our findings and the more general phenomena of auditory hypersensitivity.

We think that addresses the reviewer’s confusion. The point we are trying to make is to show that after high-frequency sensorineural hearing loss, low-frequency tones recruit stronger activity in the deafferented high-frequency zone of the tonotopic map. Recruiting more neurons into processing low-frequency tones would theoretically enhance neural population sensitivity to that stimulus, in line with the behavioral changes that we report. The point of Figure 5 is that a decoding analysis of cortical ensemble activity on single trials of tone presentation indeed shows enhanced cortical detection of 8kHz tone presentations, similar to what we observed behaviorally. Across the same range of sound intensities, neural classification of high-frequency tones is reduced on account of the threshold shift. The main point is that a neural classifier trained only on a small number of cortical neurons shows a clear parallel to the behavioral finding of auditory hypersensitivity.

Also, it seems that neural activity is unchanged after noise trauma in the "intact region" when neural activity is assessed by calcium imaging, while the activity has been found to be largely enhanced in that region when neural activity is assessed with electrophysiology (MUA and LFP). That should be mentioned in the discussion. I am not a specialist in calcium imaging but the authors may explain somewhere (briefly) the relationship between calcium responses and MUA and LFP. Is there any correlation between calcium responses (the max) and MUA response and LF amplitude?

Sorry for the confusion. We reported significant changes in the low-frequency “intact” map region as well. Figure 5C shows increased neural gain shortly after noise exposure in this region. Yes, it subsides with additional time following acoustic trauma, but it is certainly prevalent, at least initially. Also, Figure 7F shows persistent increases in spontaneous activity and neural synchrony in the low-frequency area near the deafferentation boundary.

To the reviewer’s questions about how GCaMP fluorescence compares with multiunit activity and LFP, we are not aware of any direct comparison across these measurement modalities. In an earlier paper, we compared MUA, single unit, and LFP activity to a simulation of 2-photon calcium imaging data (see Figure 9 of Guo et al., J. Neurosci 2012) but it wasn’t reported in the context of hearing loss or central gain changes.

Could the authors explain how they assess spontaneous activity from calcium imaging? What is the shape of the signal that is recorded? Is this signal similar to spontaneous LFP? Spiking activity? This is important when authors assess the synchrony between cells, and then understand what is synchronized. If the activity is globally increased we expect a "mechanic" increase in synchrony that is difficult to correct. With spiking activity, it is different (it is easier to correct, except for onset and offset responses) since the time duration of a spike is very short compared to the inter-spike interval.

Figures 3D and 7A illustrate averaged evoked and single trial spontaneous calcium transients, respectively. Imaging data were processed with Suite2p, a publicly available software package that provides a complete pipeline for processing calcium-dependent fluorescence signals collected with two-photon microscopes (Pachitariu et al., 2016; Stringer and Pachitariu 2019) that is widely used by neuroscience laboratories. Briefly, fluorescence data was processed in four stages:

1) Frame Registration: Brain movement artifacts are removed through a phase correlation process that estimates the XY offset values that bring all frames of the calcium video into register.

2) Detecting Regions of Interest: Suite2p then identifies candidate cellular regions of interests (ROIs) using a generative model with three key terms: 2a) a model of ROI activity, 2b) a set of spatially localized basis functions to model a neuropil signal that varies more gradually across space, and 2c) Gaussian measurement noise. Fitting of this model to data involves repeatedly iterating stages of ROI detection, activity extraction, and subsequent pixel re-assignment.

3) Signal Extraction and Spike Deconvolution: Suite2p then extracts a single fluorescence signal for each ROI by modelling the uncorrected fluorescence as the sum of three terms: 1) a somatic signal due to an underlying spike train, 2) a neuropil trace scaled by an ROI-specific coefficient, and 3) Gaussian noise (Stringer and Pachitariu 2019). The uncorrected fluorescence is first extracted by averaging all signals within each ROI. The neuropil trace is then computed as the average signal within an annular ring surrounding each ROI. The neuropil component is different from those identified during ROI detection, which implicitly uses pixels inside ROIs, and are not scaled by a contamination factor. Neuropil scaling coefficients and somatic fluorescence are then simultaneously estimated using an unconstrained non-negative deconvolution, using exponential kernels.

4) Cellular Identification: With a fluorescence trace assigned to each identified ROI, the final stage in the Suite2p pipeline involves identifying the subset of ROIs that correspond to neural somata. Suite2p utilizes a semi-automated approach by first labelling ROIs as cells or noncells based on various activity-dependent statistics, before a final manual curation step.

The reviewer’s question is really focused on the last part of Step 3, in which the relatively sluggish time-course of the calcium signal is deconvolved to produce sharper transients that are closely linked to spiking events. The kernel used for deconvolution in Suite 2P is derived from experiments that performed simultaneous cell-attached recordings and GCaMP6s imaging. In essence, the deconvolution kernel is a Rosetta Stone that analytically corrects for the intrinsic sluggishness of the calcium indicator to provide something akin to a ground truth assessment of spike timing. To be fair, even these ultrasensitive indicators can be less reliable for single spike events, especially with larger imaging areas (i.e., fewer pixels per somata) but this would be a constant across groups (acoustic trauma vs sham), imaging time, and cortical location so it is not a confounding influence.

DiscussionI suggest being much more cautious in the interpretation of homeostatic plasticity and behavioral data."Furthermore, to control for any changes in licking behavior due to acoustic trauma, our slope measurements are taken from functions using a sensitivity index (d'), which normalizes lick probability according to the false alarm rate determined from the delivery of catch (silent) trials. Thus, a steeper relationship between increasing tone intensity and perceptual sensitivity strongly suggests a hypersensitivity to sound following acoustic trauma (Figure 2)."I am not convinced by this conclusion. At the threshold, enhanced arousal (due to stress after noise trauma) may increase d'. I am not saying stress is necessarily enhanced in exposed animals but it is just to mention that the link between psychometric function for thresholds and loudness is not as straightforward as the authors seem to believe it is.

To address the reviewer’s request, we removed the second sentence that is quoted above and replaced it with a more cautious conclusion (Pgs. 13-14, Lns 436-442):

“To control for changes in global behavioral state due to acoustic trauma, our slope measurements are taken from functions using a sensitivity index (d’), which normalizes lick probability according to the false alarm rate determined from the delivery of catch (silent) trials. Thus, overall changes in stress, arousal, or other global behavioral states following acoustic trauma that impacted overall responsivity in catch and stimulus trials would be controlled for by the d’ measurement. However, as with reaction time, the d’ growth function is not a direct measure of loudness, per se, but instead is probably best likened to the change in stimulus salience for tones of varying intensity.”

The discussion should come back a bit more to the literature and emphasize in more detail what is consistent with the literature, what is not, what is new, etc.

We refocused the Discussion along these lines. We removed text that speculated about topics for future investigation and replaced it with text that more explicitly describes what our experiments show (or don’t show). In addition, we have added new citations on layer-dependent expression of cortical changes after acoustic trauma and behavioral indices of auditory hypersensitivity.

I suggest publishing the behavioral data elsewhere, and to focus here on the neural responses before and after the noise trauma, emphasizing clearly what is new, etc.

We appreciate that the reviewer felt that the behavioral data detracted from the more convincing aspects of the study. Overall, the reviewer’s main concern was whether this behavior was related to the loudness hyperacusis phenotype observed in clinical populations and – more specifically – whether this behavior supported strong conclusions about loudness hypersensitivity at all. As noted in the revisions above, we think these are both fair critiques and we have revised the paper to explicitly state that our study should not be interpreted as a model of hyperacusis, nor that we are directly measuring loudness.

Removing the behavioral data seemed too extreme to us. The reviewer noted that s/he “loved” the behavioral experiment that showed hypersensitivity to direct thalamocortical stimulation. Further, the analysis of the 2-photon data in Figures4-6 is predicated in no small part by the auditory hypersensitivity behavioral finding. We feel that the impact of the manuscript would be greatly reduced by the removal of the behavioral experiments but agree that a more accurate description of the behavior and a more cautious, even-handed interpretation of what the findings mean help to underscore what is new and significant in our work.

Reviewer #3 (Recommendations for the authors):Specific concerns:IntroductionThe communalities between tinnitus, hyperacusis and speech in noise impairment are unknown, and the association between those and hearing loss are known but not what happens exactly at neural level. It seems extreme to put together normal aging with ADHD, autism, or schizophrenia, and claim that tinnitus, hyperacusis, and speech in noise impairment can be involved in all those conditions.

We make no such claim. The introduction only makes the point that sensory overload (as defined by phantom percepts, hypersensitivity, or heightened distractibility) is very common in the auditory modality and is observed in a wide range of neuropsychiatric disorders. There is nothing extreme or controversial about this point from our perspective. This is a general biology journal, and we feel that it is important to begin the manuscript with an introduction that would interest readers from diverse fields before delving into the narrower topic of acoustic trauma and auditory hypersensitivity.

Line 76: we don't know yet whether neural gain adjustments following adult-onset hearing loss overshoot the mark. In fact, Koops, Renken, Lanting and van Dijk claimed that cortical tonotopic map changes in humans are larger in hearing loss than in additional tinnitus. J Neurosci. 2020 Apr 15; 40(16): 3178-3185. doi: 10.1523/JNEUROSCI.2083-19.2020

The evidence is overwhelmingly clear in animal models that adult-onset sensorineural hearing loss induces excess neural gain, particularly at the level of the auditory cortex. This has been documented in dozens of studies, many of which are cited here. Further, as the reviewer notes, the Koops 2020 study in human subjects explicitly makes the point that subjects with high-frequency hearing loss exhibit increased central gain compared to normal hearing controls. This is the same point we are making and what our animal model of hearing loss most closely approximates. We are not attempting to model or study tinnitus and make no claims to this effect. That the gain elevation is not as steep in human subjects with both hearing loss and tinnitus has no real bearing on the point we are making here, as we have no indication one way or another that the mice in our study experienced tinnitus.

Please stay consistent with the use of trauma, acoustic trauma, sound overexposure, and high-frequency sound exposure throughout the manuscript.

Thanks for this suggestion. We agree that the wording was confusing on this point. We consistently use “trauma” and “sham” in the figures, where space is often tight. In the text, we use “acoustic trauma” and “sham” to refer to the sound exposure conditions. The revised manuscript has no occurrences of “sound overexposure” or “high-frequency sound exposure”.

ResultsA clarification about the percentage of animals that develop loudness hypersensitivity after sound overexposure is necessary. As previously mentioned hyperacusis and tinnitus are not happening to all animal models of sensorineural hearing loss overexposed to high-frequency sounds. How the authors distinguish between animals with sensorineural hearing loss and animals with associated tinnitus and/or hyperacusis?

As mentioned in the response to the public reviews, we appreciate the reviewers’ comments and acknowledge the limitations in relating our behavioral measurement to hyperacusis. The revised figure 2 features a new panel showing the fold change in d’ slope relative to baseline over the post-exposure period for each individual mouse. As described in the response to the public reviews, we noted a variable degree of increased sensitivity to mid-frequency tones in all mice after acoustic trauma but in only one sham-exposed mouse.

Perceptual hypersensitivity following noise-induced high-frequency sensorineural hearing loss.Please state more precisely the differences between hyperacusis and hypersensitivity to loud sounds.

The revised text clarifies the relationship between our findings and the clinical phenomena of loudness recruitment and hyperacusis.

For loudness recruitment, we have added the following text on Pg. 6, Lns 148-154.

“Persons with sensorineural hearing loss commonly report loudness recruitment, a disproportionately steep growth of loudness with increasing sound level, that has been accounted for by altered basilar membrane mechanics after OHC damage (Oxenham and Bacon, 2003). However, loudness recruitment is most pronounced for frequencies within the range of hearing loss and at intensities well above sensation level (Buus and Florentine, 2002; Moore, 2004). Here, we observed steeper growth of auditory sensitivity for a spared frequency and at intensities close to sensation level, neither of which would be expected of a purely peripheral origin.”

For hyperacusis and behavioral measures of loudness, we made text changes throughout the manuscript as well as revising the following paragraph, Pg. 14, Lns 448-466.

“While the findings presented here support an association between sensorineural peripheral injury, excess cortical gain, and behavioral hypersensitivity, they should not be interpreted as providing strong evidence for these factors in clinical conditions such as tinnitus or hyperacusis. Our data have nothing to say about tinnitus one way or the other, simply because we never studied a behavior that would indicate phantom sound perception. If anything, one might expect that mice experiencing a chronic phantom sound corresponding in frequency to the region of steeply sloping hearing loss would instead exhibit an increase in false alarms on high-frequency detection blocks after acoustic trauma, but this was not something we observed. Hyperacusis describes a spectrum of aversive auditory qualities including increased perceived loudness of moderate intensity sounds, a decrease in loudness tolerance, discomfort, pain, and even fear of sounds (Pienkowski et al., 2014a). The affective components of hyperacusis are more challenging to index in animals, particularly using head-fixed behaviors, though progress is being made with active avoidance paradigms in freely moving animals (Manohar et al., 2017). Our noise-induced high-frequency sensorineural hearing loss and Go-NoGo operant detection behavior were not designed to model hyperacusis. Hearing loss is not strongly associated with hyperacusis, where many individuals have normal hearing or have a pattern of mild hearing loss that does not correspond to the frequency dependence of their auditory sensitivity (Sheldrake et al., 2015). While the excess central gain and behavioral hypersensitivity we describe here may be related to the sensory component of hyperacusis, this connection is tentative because it was elicited by acoustic trauma and because the detection behavior provides a measure of stimulus salience, but not the perceptual quality of loudness, per se.”

Explain why only wave 1 was used to explore the auditory thresholds in ABRs when wave 5, for example, would be easy to identify.

A vertical electrode montage (pinna-vertex) is more sensitive the electrical dipole generated by central auditory structures downstream of the cochlear nucleus that are the primary contributors to waves 3-5. Instead, we use a horizontal (pinna-pinna) electrode montage due to the surgical procedure preventing electrode placement on top of the head, which is sensitive to the dipole generated by synchronized activity of the spiral ganglion neurons (wave 1) and cochlear nucleus (wave 2) but waves 3-5 are not easily discernable. This has been explained in many of our previous publications (e.g., Chambers et al., Neuron 2016) and also in seminal reports (Galbraith et al., J Neurosci Methods 2006). Perhaps most importantly, later ABR waves can reflect central gain changes, while the main point of Figure 1 is to provide a comprehensive characterization of the peripheral sensorineural damage at the level of stereocilia damage, hair cell death, primary afferent synapse degeneration, otoacoustic emissions, and peripheral neural response. Wave 1 is most suitable component of the ABR for this purpose.

We revised the Methods to clarify this point on Pg. 20 Lns 689-691,

“To highlight the peripheral neural contribution to the ABR, subdermal electrodes were positioned in a horizontal (pinna-to-pinna) montage (Galbraith et al., 2006; Melcher et al., 1996).”

It is not clear how far the animals were tested and the survival time of the animals between the hearing loss triggering and the histology showing loss of synaptic contacts in IHCs in the base of the cochlea and death of OHC in the base with stereocilia changes in base and mid coils. In the different figures, there are different days and groups of days depending on the figure. Please clarify.

The preparation of cochlear materials for the histopathology analysis was performed 21 days after exposure. This point has been clarified by the addition of Supplementary File 1, which provides the timing of each procedure across all mice and through changes to the text on Pg. 5, Lns 113-114:

“Post-mortem cochlear histopathology performed 21 days after noise exposure suggested an anatomical substrate for cochlear function changes due to acoustic trauma.”

Figure 1 panels J, K. Change the colours to make them more different. Why there is a threshold elevation at 8 kHz in the first two days?

We adjusted the colors in Figure 1J-K so that they are more easily discriminable. The temporary threshold elevation at 8 kHz in the first 24 hours after acoustic trauma reflects TTS (temporary threshold shift). TTS is typically observed over a wider frequency range than permanent threshold shift.

Behavioural hypersensitivity to cochlear lesion edge frequencies and direct auditory thalamocortical activation after SNHL.Would it be possible that the differential distribution of damage between the IHCs and OHCs (only base versus base and middle coil) could explain the changes in ABRs, DPOAE and behavioural thresholds?

Yes, certainly. Behavioral thresholds (Figure 1J-K) exhibited lasting elevation at 32kHz but not 8 kHz. That is easily explainable by observation of OHC and synaptic degeneration at the 32 kHz region of the cochlea but not the 8 kHz region. ABR and DPOAE threshold elevation is <10dB up to 16 kHz but then increases to 30-40dB above 16 kHz. ABR and DPOAE thresholds are less sensitive to afferent lesions (IHC or synaptopathy) and are mostly reflective of OHC damage. OHC elimination was observed at the extreme base (Figure 1E), which can account for some of ABR and DPOAE threshold elevation, but the changes in the 22-45 kHz range arises from more subtle damage to OHCs. Some OHC stereocilia damage at lower regions of the cochlear frequency map can be observed at the light microscopic level (Figure 1G) but resolving the physical substrate of the ABR and DPOAE threshold elevation at lower frequencies requires techniques that can go below the diffraction limit (e.g., electron microscopy).

Please clarify why in the combined optogenetic (ChR2) and auditory stimulation, only a high frequency centred at 32 KHz. Is the optogenetic stimulation affecting all frequency ranges in vMGN or only high frequency?

Trials of thalamocortical stimulation were interleaved with high-frequency acoustic stimulation as a positive control; we wanted to show that enhanced sensitivity to central auditory stimulation was accompanied by decreased sensitivity to sound frequencies corresponding to the high frequency cochlear damage. Further, by keeping the 32kHz trials at 50% in both versions of the behavioral detection task (where the other 50% were more easily detected low-frequency tones or optogenetic stimulation), we maintained the overall rates of miss trials and reward between the two variants of the behavioral task.

We have no insight into the frequency tuning of the virally transduced MGB neurons. There is no reason to believe our injections targeted MGB neurons tuned to any particular range of sound frequencies.

In Figure 2, panel E it seems that the psychometric curve in the baseline condition for the hearing loss animals is shifted to the right compared with the baseline in the sham condition. It would be easier to compare them if they are placed in the same panel.

Thanks, the reviewer’s critique helped us improve this figure. We selected a more representative trauma mouse for Figure 2E and revised Figure 2F to indicate which lines correspond to the trauma and sham exemplars shown at left. For the example mice, it was important to show the changes within a mouse that occur (or fail to occur) after noise exposure relative to baseline testing. The quantification of the optogenetic behavior is also based on changes within an animal in the post-exposure period relative to baseline because the particular growth functions vary from mouse to mouse based on fiber placement, transduced neurons, and the optical interface between the fiber and the transduced neurons. For this reason, we determined the most informative contrast was to include data from different behavioral sessions within a mouse rather than a single session for different mice.

Chronic imaging in A1 reveals tonotopic remapping and dynamic spatiotemporal adjustments in neural response gain after acoustic traumaThe authors mentioned that tonotopic remapping was limited to the deafferented zone where the proportion of layer 2/3 pyramidal neurons preferentially tuned to lesion edge frequencies is more than doubled following acoustic trauma without any systematic change in response. Please specify the lack of systematic change.

Pg. 7 Lns 200-203 state,

“Analysis of all tone-responsive PyrNs (n = 1,749 in 4 trauma mice; n = 1,748 in 4 sham mice) demonstrated that tonotopic remapping was limited to the deafferented zone (Figure 3G), where the percentage of L2/3 PyrNs preferentially tuned to lesion edge frequencies more than doubled following acoustic trauma (Figure 3H) without any systematic change in response threshold (Figure 3I).”

We were specifically referring to the minimum response threshold. This analysis was inspired by Dexter Irvine’s publications on cortical plasticity after hearing loss, which showed that tonotopic remapping versus residual tuning can be distinguished by whether the preferred frequency changes without an elevation in response threshold.

The identification of the frequency edge is convincing in the widefield but less convincing at the neural level. How the deafferentation boundary was established? Please specific the rational for the use of those four specific frequencies (32, 5.7, 8, 11.3)

On lines 783-787, we state,

“To delineate the intact and deafferented regions of the imaging field, a support vector machine (SVM) was calculated for each mouse using BF’s determined from the first day of imaging prior to noise exposure. The SVM deafferentation boundary categorized physical space (intact: BF < 16 kHz, deafferented: BF ≥ 16 kHz) and its physical location was then imposed on all successive imaging sessions after alignment of fields-of-view from all imaging sessions.”

The four frequencies were selected such that we had one frequency corresponding to the region of sensorineural hearing loss (32k), one frequency far from the cochlear damage (5.7kHz), one edge frequency corresponding to the behavioral experiments (8kHz), and one edge frequency bordering the low-frequency edge of ABR and DPOAE threshold elevation (11.3 kHz).

This is stated on Pg. 7, Lns 214-217:

“Next, we expanded the neural gain analysis in sham and trauma mice to four stimulus frequencies: a high-frequency tone aligned to the damaged cochlear region (32 kHz), a spared low-frequency tone far from the cochlear lesion (5.7 kHz), and two spared mid-frequency tones near the edge of the cochlear lesion (8 and 11.3 kHz).”

Cortical hyperresponsivity and increased gain mirrors behavioural hypersensitivity to spared low-frequency inputs.This section is not particularly convincing and figure 5 is difficult to follow. For example, in panel D it is impossible to distinguish the two different types if sound or silence. Only panel F is clear and easy to follow.

One of the main contributions in this study is that it provides the means to study many key variables related to cortical plasticity after hearing loss that have traditionally only been studied in isolation. For example, 1) the cell’s tonotopic map location relative to zone of cochlear damage is known to be an important determinate of plasticity, 2) as is the length of time separating the measurement and the cochlear injury, as well as 3) the frequency of the stimulus used to elicit a sensory response relative to the region of cochlear damage. Of course, each of these features depend on 4) whether the noise exposure caused sensorineural cochlear damage or was innocuous.

Here, we measure each of the dimensions in single neurons and document a 4-way interaction; in other words, the degree of central gain depends on all four factors. For example, gain changes relative to sham controls are weak with low-frequency tone in the low-frequency portion of the map more than a few days after acoustic trauma. Conversely, excess gain relative to sham controls is greatest for lesion edge frequencies (11.3k) in the deafferented region of the tonotopic map several days after acoustic trauma. Dropping any of these three dimensions diminishes the message of the project, hence all must be shown. As such, a certain degree of complexity is unavoidable in these figures as there are four independent variables. We can think of no more convincing and simple way of plotting the gain changes across these four variables than the line plots in Figure 5A-C. However, we tried to address the reviewer’s comment by making text changes that seek to simplify an inherently complex result (Pg. 8, Lns 220-226):

“This analysis identified several clear results: i) a strong initial uptick in neural gain measured in both topographic regions following trauma; ii) persistent (lasting greater than 1 week) increases in neural gain were observed only for spared mid-frequency tones in deafferented cortical regions; iii) no significant changes in neural gain were observed in sham-exposed mice (Figure 5C). Thus, excess central gain reflected the interaction of four factors: 1) whether the initial sound exposure induced SNHL, 2) where within the cortical frequency map the cell is located, 3) when relative to exposure the measurement is made, and 4) the proximity of the stimulus test frequency to the cochlear lesion.”

For Figure 5D-F, we relate neural response gain to the behavioral result. This was accomplished by constructing a linear SVM decoder where the input is the population activity in auditory cortex and the decoder is trained to distinguish between trials of sound and silence. Before applying the SVM decoder, we first reduce the dimensionality of the cortical ensemble response using PCA. Figure 5D provides a visualization of the SVM decoding by showing the first 2 principal components for single trials of silence and sound. The reviewer commented that it is impossible to distinguish the sound and silence trials in some conditions (e.g., the first few columns) and this is exactly the point. The cortical decoding model struggles to detect low-intensity tones and the overlap in the data is intended to convey that point. Conversely, at high intensities, particularly after acoustic trauma the blue and gray data points are well separated. The reviewer mentions that panel F is difficult to follow, though s/he did not mention why. It simply shows the change in cortical detection for low and high frequency tones before and after acoustic trauma. Detection accuracy improves for low-frequency tones and becomes worse for high frequency tones, paralleling the behavioral results. We tried to address the reviewer’s critique but determined that the current organization of Figure 5A-C is optimal to convey the 4-way interaction between exposure type, map location, stimulus frequency and post-exposure day. Further Figure 5D-F is the optimal way to visualize and convey the findings from the single trial cortical ensemble decoding analysis.

Tracking changes in activity and local synchrony in individual PyrNs over several weeks.Explain the differences across neurons in the probability to be tracked just for one day or for 15 session. Please explore the possibility that the baseline properties of the neurons impact their tractability in chronic imaging

Like extracellular recordings, 2-photon calcium imaging is biased towards excluding inactive or very sparsely active neurons from the analysis. Cells are “lost” from the chronic tracking for two reasons: 1) they become virtually quiescent and therefore are “invisible” to the regions of interest (ROI) identification process; 2) the geometry of the cell (i.e., the somatic ROI shape) or the geometric separation from other cells change to a degree that we cannot be confident it is the same cell.

We implemented a relatively rigorous process for establishing ROI tracking confidence that is described in Figure 6 —figure supplement 1 and related text. We observed a constant turnover of ROIs (new ROIs gained, prior ROIs lost) that likely reflects the inherent difficulty of returning to exactly the same XYZ coordinates over time. In addition, we noted a large turnover in lost/added ROIs only after acoustic trauma, only during the 48 hours after exposure, and only near the deafferentation boundary in the cortical map. The two most likely explanations for the large uptick in cell turnover is 1) some active neurons become virtually quiescent and vice versa, and/or 2) the topology of the cortex changes in newly deafferented areas, disrupting our ability to track ROIs based on their appearance and relation to other ROIs. To this latter point, prior studies have documented a degradation of the extracellular matrix proteins (e.g., doi: 10.1016/j.heares.2017.04.015) and uptick in neurite motility (doi: 10.1016/j.neuron.2011.12.038) that accompany suden hearing loss. In theory, this could destabilize the physical geometry of pyramidal neuron ensembles near the deafferentation boundary in the first few days after acoustic trauma.

We revised Pg. 9, Lns 257-267 to convey these points with greater clarity,

“Interestingly, we noted that new active PyrNs appeared immediately following noise exposure, while PyrNs tracked throughout the baseline imaging sessions disappeared (Figure 6 —figure supplement 1C). Because the degree of turnover was far less after sham exposure, because the appearance or disappearance of PyrNs after acoustic trauma was concentrated near the deafferentation boundary (Figure 6 —figure supplement 1D), and because approximately 75% of PyrN disappearance occurred within the 48 hours after acoustic trauma (Figure 6 —figure supplement 1E), we concluded that increased PyrN turnover in the imaging field must be related to cortical changes arising indirectly from the cochlear SNHL. Possibilities include large and heterogenous state shifts in activity (from virtually quiescent to active or vice versa) as well as changes in the physical topology of the cortex due to degradation of extracellular matrix proteins and increased neurite motility after sudden hearing loss (Nguyen et al., 2017; Tschida and Mooney, 2012).”

To the reviewer’s comment about baseline properties, we would refer them to the multivariate model described in Figure 8. Here, we use the diversity of baseline properties to account for diverse functional outcomes after hearing loss. This analysis ensures that our imaging approach includes cells with wide-ranging response properties at baseline and can return to them several days later. So, apart from near-complete quiescence, we cannot identify a scenario in which the baseline properties of the neuron impact its tractability for chronic imaging, though they certainly influence the response to hearing loss, as detailed in Figure 8.

The authors mention that following noise overexposure active pyramidal neurons disappeared from the deafferented area, with a permanent increase in response to 8 kHz tome in the deafferented area but not in the low-frequency area. This increase was first for all intensity levels and later only above 50 dB SPL. When this changes happen? After 3 days? When is the beginning of the 2nd stage? Could the authors distinguish between an acute and a chronic phase?

Taking the reviewer’s questions one at a time, changes in the response to an 8 kHz tone over post-exposure day is provided in Figure 6D. The short-term gain increase in low-frequency map areas is strongest hours after the trauma (D0) and is gone by day 3. Likewise, as in shown now in Figure 6 – Supplement 1E, about 70% of the cells that were ‘lost’ disappeared within the first 48 hours after trauma. By contrast, among the relatively small number of PyrNs that were lost in the sham exposure group, only about 20% were lost within the first 48 hours after the innocuous noise exposure.

Taken together, yes, the reviewer is correct in observing a period of acute changes in tone-responsiveness following noise exposure that are followed by stable, sustained hyperresponsivity, as shown in Figures 4-6. These results highlight Days 0-2 as acute and highly dynamic, while Days 3+ show far more stable changes in auditory cortex activity. This timescale is also reflected in both the appearance and disappearance of active pyramidal neurons, where on days 0-2 following noise exposure these changes are the most prominent.

We revised the text on Pg. 10, Lns 304-309 to make this clearer to the reader:

“Taken together, these observations also suggest that plasticity following acoustic trauma is organized into two phases: a dynamic phase during the first 48-72 hours after noise exposure involving topographically widespread hyper-correlation, hyper-responsivity to mid-frequency sounds in the intact map regions, and large-scale turnover in PyrN stability around the deafferentation map boundary, followed by a stable phase beginning 3 days after exposure where gain is increased for spared tone frequencies in deafferented map regions.”

Predicting the degree of excess central gain after trauma in individual PyrNs based on their pre-exposure response featuresPlease expand your explanation about potential heterogeneity. I am not certain whether the authors talk about two different groups of pyramidal neurons or a gradient between low to high spontaneous activity at baseline.

In Figure 8D, we show that there is a gradient over which baseline spontaneous activity may have a relationship with post-exposure changes in response gain. In our linear model, most predictor variables are continuous, which is explained in the Figure 8 legend and the Methods, and so we expect that there is a continuous relationship over which each of these variables relates to the post-exposure gain. Given the results of the linear model, and to offer a reasonable and readable interpretation, in Figure 8G we provide a graphical summary of the model results. Here, we highlight two ‘classes’ of neurons (those with stable gain and those with increased gain), but we also include graphics indicating a gradient of influence for each predictor variable, as this is how linear models work for continuous variables.

Again here the temporal limit is different and they use 3.5 days. Please explain the rationale of that and use always the same time points with justification.I am curious to know what will be the result of those functions about central gain if other frequencies were used.

We used 3-5 days post-exposure because the initial phase of acute reorganization and rapid neural dropout has ended and the expression of plasticity is equivalent to what is observed at longer post-exposure observation periods. Thus, by 3-5 days after trauma, the reorganization has stabilized while affording us the greatest number of chronically tracked PyrNs to include in our model (which helps to avoid overfitting).

Multivariate linear regression is limited when it comes to including separate imaging sessions in the model. There are more complex modeling approaches that can account for multiple timescales (e.g., tensor decomposition) but we were not in a position to attempt those modeling approaches, so we determined that selecting a single post-exposure period of 3-5 days was the most logical and informative way to go. The reviewer makes a good point that the rationale for this decision was not made clear and we have revised the text on Pgs. 10-11 Lns 327-329 to address this point:

“To avoid overfitting the regression model, we selected a single timepoint – 3-5 days after noise exposure – that corresponded to the stable phase of reorganization after acoustic trauma, while maximizing our sample size of chronically tracked PyrNs.”

As for modeling the changes in the same neurons at other stimulus frequencies, our sample size limited how many terms we could add into the model without overfitting and without running into problems with multiple comparisons from the same sample. For these reasons, we limited our model to a single frequency and determined that 8kHz was the best choice as it was the frequency used for behavioral testing (Figure 2) as well as the neural classifier (Figure 5).

DiscussionPlease state the differences between the hearing loss in the animal model by acute sound overexposure from the majority of ageing sensorineural hearing loss (presbyacusis) where the changes are more gradual.Excess central gain is the hypothesized neural substrate for loudness recruitment and generalised hyperacusis, but can the authors think about other potential causes?

Both presbycusis and noise-induced hearing loss feature afferent synapse loss, high-frequency threshold elevation, and high-frequency outer hair cell stereocilia pathology. Animal models for age-related hearing loss generally identify additional breakdown in the tight cellular junctions and ion transporter proteins that power the endocochlear potential that provides additional low-frequency threshold shift that is not typically found in acoustic trauma.

In terms of the reviewer’s questions, yes, the revised text now mentions that loudness recruitment can reflect altered basilar membrane mechanics on Pg. 6, Lns 148-150: Persons with sensorineural hearing loss commonly report loudness recruitment, a disproportionately steep growth of loudness with increasing sound level, that has been accounted for by altered basilar membrane mechanics after OHC damage (Oxenham and Bacon, 2003).

Other underlying contributions to excess central gain and associated auditory perceptual hypersensitivity disorders is discussed on Pgs. 3-4 Lns 58-89 and Pgs. 12-13 Lns 368-426.

Line 353. Typo long spaces in the same line.

Fixed.

Why and what for pyramidal neurons in layers 2-3 with low spontaneous activity and greater responses to low-intensity tones show more stable gain control after trauma? I do not understand what "stochastic variations in inhibitory tone between different microcircuit milieus" means. Is it about the different types and functions of inhibitory interneurons in the cortex? Can neuromodulators such as ACh play a role too?

The design of our study limits our insight into questions related to “why and what for”. However, the Discussion speculates on this point that L2/3 pyramidal neurons embedded into microcircuits that impose stronger constitutive inhibition would be associated with more non-monotonic intensity growth functions and lower spontaneous firing rates. Conversely, microcircuit milieus that impose weaker feedforward inhibition onto pyramidal neurons might be reflected in higher spontaneous rates and more linear intensity response growth functions.

This point is made on Pgs. 12-13, Lns 400-405.

“In particular, PyrNs with low spontaneous activity rates and non-monotonic encoding of sound intensity – features associated with stronger intracortical inhibition (Tan et al., 2007; Wu et al., 2006) – showed more stable gain control after trauma. Conversely, initially high spontaneous activity, hyper-correlation, and gradual monotonic intensity growth functions could be reflective of weaker intracortical inhibitory tone and were features that predicted non-homeostatic excess gain after acoustic trauma.”

Can the different types of neurons be distinguished? Again, what happens when the animals only have hearing loss without tinnitus or hyperacusis? Should be possible that the accumulation of one or more than one symptom after hearing loss depends on the proportion of active cells on the baseline?

These are all interesting questions that cannot be directly addressed by our experiments. We speculated that baseline response features associated with non-homeostatic excess gain (high spontaneous firing rate and gradual monotonic growth of response to 8 kHz tones) might be reflective of weak local inhibitory tone, which in turn would be predictive of poorly regulated gain after acoustic trauma. To the reviewers, question, other than the certainty of knowing that they are L2/3 excitatory pyramidal neurons, we don’t have any evidence one way or the other to say how the pyramidal neurons that show excess gain versus stable gain are intrinsically different (i.e., different morphology, biophysics, genetics, etc.). We have a lot of evidence to say that we can account for 40% of the variance in how they response to acoustic trauma based on where the neurons are located in the tonotopic map and other functional characteristics that are presumably regulated by local and random (i.e., stochastic) variations in local circuit motifs.

The reviewer’s questions about types of inhibitory interneurons, a possible role of neuromodulators, and differences in animals with hearing loss that either do (or do not) have behavioral evidence of tinnitus and hyperacusis are interesting ideas for casual discussion but are not directly related to the experiments that we performed, so we opted not to address these points through text revision.

The authors mention the startle reflex as an indicator of hypersensitivity but usually, animals show a high degree of adaptation to it. Can the sensory component of the hypersensitivity show habituation or sensitization?

From our behavioral measurements, we see stable mouse performance over time in both sound and optogenetic detection, both before and after noise/sham exposure. Given this observation and the consistent trend within groups, we do not see evidence of habituation or sensitization from our measurements. This consistency may come from the difficulty of our tasks. In both procedures, we use stimulus intensities where the mice are operating near their perceptual threshold, which allows for the mice to stay engaged in the task for long periods of time. This point is illustrated by the example adaptive track show in Figure 1I, the consistent behavioral thresholds over days shown in Figure 1K, and the stable gain elevation observed after acoustic trauma in Figures2B and 2F.

Line 439, again many spaces in the sentence.

Fixed

Line 448. Check typos in references.

Fixed

Lines 466-468 needs a reference. Again here, the authors are not identifying tinnitus or even hyperacusis or misophonia?

We have added references to this sentence (Pg. 15, Lns 492-493).

MethodsGive reasons about the time of 9 weeks post-natal when sound overexposure (9 weeks post-natal) and why was done in the morning that corresponds with the period of higher sleep pressure for the mice.

We wanted to model noise-induced high-frequency sensorineural hearing loss in adults. The peripheral and central auditory system is mature by 9 weeks. That is why we chose 9 weeks. Noise-induced cochlear injury is regulated by the circadian clock. Our main intent was to make the exposure time consistent across mice. But we noted prior work showing that the temporary component of the threshold shift was less extreme and less variable for noise exposures occurring during the day, which suggested that morning was a good choice for the exposure time (see Meltser et al., Current Biology 2014, http://dx.doi.org/10.1016/j.cub.2014.01.047).

To address the reviewer’s comment we have revised Ln. 526-528 of the Methods to read,

“Noise exposure occurred at 9 weeks postnatal and was timed to occur in the morning, when the temporary component of the threshold shift is less extreme and variable (Meltser et al., 2014).”

It is difficult to follow the different animals in the experimental design. Please use a table

To address the reviewer’s request, we added Supplementary File 1, which provides timing for all experimental animals, now referenced on Pg. 16, Ln 539.

Please specify if the whole ventral division of MGB was targeted with the fibre optic or only the high-frequency region. Was the fibre optic present but inactive in MGB for the control experiments?

There is no way to restrict the cone of light or the virus expression only to the high frequency BF/CF region of the MGB tonotopic map using the approach described in the Methods. The control group shown in Figure 2 is a sham-exposed control. The treatment of these mice differs only in the sound pressure level of the noise exposure. The preparation for the optogenetic experiment is the same including the virus injection and fiber implant, as evidenced by Figure 2E, which shows the psychometric detection function for direct thalamocortical activation in a control mouse.

Line 571-3. The range of sound levels and laser powers were tailored to each mouse prior to noise exposure to ensure equivalent sampling of sound and laser perceptual growth functions. Please expand the explanation by giving further information about differences across cases.

To study changes in the perceptual salience of near-threshold sounds, we used adaptive staircasing (i.e., Method of Limits) to present sounds just above and below threshold (Figure 2A-B). For the combined optogenetic and acoustic task, we wanted to show changes in performance for a fixed set of stimuli before and after acoustic trauma or sham exposure (i.e., Method of Constant Stimuli). Subtle differences in the amount of opsin expression, the position of the fiber relative to the transduced cells, and the optical properties of the fiber-tissue interface required that we find the range of laser powers and 32 kHz SPLs that spanned the dynamic range of the mouse – from undetectable to detectable with near certainty. Once we had found this range for each mouse, we then proceeded to use these same 5 laser levels and sound levels for all subsequent testing sessions after sound exposure.

We thank the reviewer for pointing out that these details were missing from the Methods. We have revised the text on Pgs. 18-19, Lns. 614-625 to read:

“For testing, randomized interleaved blocks of either noise or laser stimulation were presented at a fixed range of levels. The range of sound levels and laser powers were individually tailored prior to noise/sham exposure to ensure equivalent sampling of sound and laser perceptual growth functions, and then these fixed values were used for all post-exposure testing sessions. Tailoring was accomplished by first identifying the lowest laser power and 32kHz sound level that produced at least 95% hit rates (operationally defined as “max”). These sound levels and laser powers were then presented alongside four attenuated levels relative to the maximum as well as no-stimulus catch trials in each mouse on every session. Psychometric functions were fit using binary logistic regression, and threshold was defined as the point where detection crossed 71% correct, which is the closest approximation to the threshold point identified with the 2-up, 1-down staircasing procedure described above. Runs were rejected for further analysis if the false alarm rate of the mouse was above 30%, and again this resulted in the exclusion of <5% of sessions.”

Line 584-585. It is bizarre using a scalpel to do a craniotomy instead of a dental drill.

The mouse skull is very thin. Drilling can introduce vibration and bruising. We have been performing mouse craniotomies with a scalpel for nearly 15 years, as described in ~40 publications from our lab. In our hands, neural responses are superior to what we get with a dental drill, so it does not seem bizarre to us.